

# Presentation of the EURODELTA III inter-comparison exercise - Evaluation of the chemistry transport models performance on criteria pollutants and joint analysis with meteorology

B. Bessagnet[1], G. Pirovano[2], M. Mircea[3], C. Cuvelier[4], A. Aulinger[5], G. Calori[6], G. Ciarelli[7], A. Manders[8], R. Stern[9], S. Tsyro[10], M. García Vivanco[11], P. Thunis[12], M.-T. Pay[13], A. Colette[1], F. Couvidat[1], F. Meleux[1], L. Rouïl[1], A. Ung[1], S. Aksoyoglu[7], J. M. Baldasano[13], J. Bieser[5], G. Briganti[3], A. Cappelletti[3], M. D'Isidoro[3], S. Finardi[6], R. Kranenburg[8], C. Silibello[6], C. Carnevale[14], W. Aas[15], J.-C. Dupont[16], H. Fagerli[10], L. Gonzalez[17], L. Menut[18], A. S. H. Prévôt[7], P. Roberts[17], L. White[19]

[1]INERIS, National Institute for Industrial Environment and Risks, Parc Technologique ALATA, F-60550 Verneuil-en-Halatte, France
[2]RSE S.p.A., via Rubattino 54, 20134 Milano, Italy
[3]ENEA, Italian National Agency for New Technologies, Energy and Sustainable Economic Development (ENEA), Via Martiri di Monte Sole 4, 40129 Bologna, Italy
[4]Ex European Commission, Joint Research Centre JRC Institute for Environment and Sustainability I-21020 Ispra (Va), Italy
[5]HZG, Helmholtz-Zentrum Geesthacht, Institute for Coastal Research, Max-Planck-Straße 1, 21502 Geesthacht, Germany
[6]ARIANET Srl, Via Gilino n.9 20128, Milano, Italy
[7]PSI, LAC, Paul Scherrer Institute, 5232 Villigen PSI, Switzerland
[8]TNO, Dept. Climate, Air and Sustainability, P.O. Box 80015, 3508 TA Utrecht, The Netherlands
[9]Freie Universität Berlin, Institut für Meteorologie Troposphärische Umweltforschung Carl-Heinrich-Becker Weg 6-10, D-12165 Berlin, Germany
[10]Climate Modelling and Air Pollution Division, Research and Development Department, Norwegian Meteorological Institute (MET Norway) P.O. Box 43, Blindern, N-0313 Oslo, Norway
[11]CIEMAT, Atmospheric Pollution Unit, Avda. Complutense, 22, 28040 Madrid, Spain
[12]European Commission, Joint Research Centre JRC Institute for Environment and Sustainability I-21020 Ispra (Va), Italy
[13]BSC, Barcelona Supercomputing Center, Centro Nacional de Supercomputación, Nexus II Building, Jordi Girona, 29, 08034 Barcelona, Spain
[14]Department of Electronics for the Automation, University of Brescia, via Branze 38, I 25123 Brescia, Italy
[15]Norwegian Institute for Air Research (NILU), Box 100, 2027 Kjeller, Norway
[16]Institut Pierre-Simon Laplace, CNRS-Ecole Polytechnique, 91128 Palaiseau, Paris, France
[17]CONCAWE, Boulevard du Souverain 165, B-1160 Brussels, Belgium
[18]Laboratoire de Météorologie Dynamique, École Polytechnique, ENS, UPMC, CNRS, Institut Pierre-Simon Laplace, 91128 Palaiseau, France
[19]AERIS EUROPE Ltd., Strouds Church Lane West Sussex RH17 7AY, United Kingdom

*Correspondence to*: B. Bessagnet (bertrand.bessagnet@ineris.fr)

**Abstract.** The EURODELTA III exercise allows a very comprehensive inter-comparison and evaluation of chemistry transport models performance. Participating models were applied over four different one month period, within a rather limited number of years (from June 2006 to March 2009) thus allowing evaluating the influence of different meteorological conditions on model performance. The exercise was performed under strict requirements concerning the input data. As a
consequence, there were very limited differences in the models set up, representing a sort of sensitivity analysis to several aspects of the modelling chains. The models were evaluated mainly on background stations. Even if the meteorology was





prescribed, some variables like the planetary boundary layer (PBL) height, the vertical diffusion coefficient are diagnosed in the model pre-processors and explain the spread of models results. For ozone, this study shows the importance of boundary conditions on model calculations and then on the regime of the gas and particle chemistry. The worst performances are observed for sulphur dioxide concentrations that are poorly captured by the models. The performances of models are rather

good very similar for the nitrogen dioxide. On average, the models provide a rather good picture of the particulate matter (PM) concentrations over Europe even if the highest concentrations are underestimated. For the PM, the mean diurnal cycles show a general tendency to overestimate the effect of the PBL height rise while the afternoon chemistry (formation of secondary species) is certainly underestimated, PM observations show very flat diurnal profiles whatever the season. In general the day time PBL height is underestimated by all models, the largest variability of predicted PBL is observed over

the ocean and seas. More generally, in most cases model performances are more influenced by the model setup than the season. The temporal evolution of wind speed is most responsible of model skilfulness in reproducing the daily variability of pollutant concentrations (*e.g.* the development of peak episodes), while the reconstruction of the PBL diurnal cycle seems more influencing in driving the corresponding pollutant diurnal cycle and hence the presence of systematic positive and negative biases detectable on daily basis.

## 1    Introduction

The ongoing project EURODELTA has very successfully extended the European Air Quality Modelling capability by providing a forum in which modelling teams could share experiences in simulating technically interesting and policy relevant problems. The joint exercises contribute to further improve modelling, techniques as well as to quantify and understand the sources of calculation uncertainty. EURODELTA is now an activity contributing to the scientific work of the

UNECE (United Nations Economic Commission for Europe) Task Force on Measurement and Modelling (TFMM) under the Convention on Long-range Transboundary Air Pollution (CLRTAP). The TFMM was established in 2000 to offer a forum to the Parties, the EMEP (European Monitoring and Evaluation Programme) centres and other international organizations for scientific discussions to evaluate measurements and modelling and to further develop working methods and tools. In that context, the Gothenburg Protocol signed in 1999 is a multi-pollutant protocol of the Convention designed to reduce

acidification, eutrophication and ground-level ozone by setting emissions ceilings for sulphur dioxide, nitrogen oxides, volatile organic compounds, fine particulate matter and ammonia.

In 2004, EURODELTA I (van Loon *et al.*, 2007) examined the common performance of the chemistry transport models (CTM) in predicting recent (2000) and future (2020) air quality in Europe using the concept of a model ensemble to measure robustness of predictions. The spread of predictions about the ensemble gave a measure of uncertainty for each predicted

value. In a 2020 world the effect of making emission reductions for key pollutants of NOx (nitrogen dioxide), $SO_2$ (sulphur dioxide), VOC (Volatile Organic Compound), PM (particulate matter) and $NH_3$ (ammonia) independently in France, Germany and Italy, and of NOx and SOx in sea areas, was investigated. Source-receptor relationships used in integrated



assessment (IA) modelling were derived for all the models and compared to assess how model choice might affect this key input. EURODELTA II (Thunis *et al.*, 2008) was built on this project by taking a closer look at how the different models represent the effect on pollutant impacts on a European scale by applying emission reductions to individual emission sectors. In the recent literature, several inter-comparison and evaluation exercises are reported for PM models : McKeen *et al.* (2007), van Loon *et al.*, (2007), Vautard *et al.* (2007), Hayami *et al.* (2008), Stern *et al.* (2008), Smyth *et al.* (2009), Vautard *et al.* (2009), Solazzo *et al.* (2012), Pernigotti, *et al.* (2013). Most of these model inter-comparison exercises were performed at the regional scale with chemistry transport models. In one of the most recent exercise, AQMEII (Solazzo *et al.*, 2012), models clearly tend to underestimate $PM_{10}$ background concentrations in US and EU regions. Model results for $PM_{2.5}$ concentrations showed better performances but large uncertainty remained certainly due to the simulation of secondary organic aerosols.

The new EURODELTA III (ED-III) exercise was promoted to exploit and interpret the EMEP intensive measurements by making modelling analysis of the campaigns to re-examine the performance of chemistry transport models. The ED-III exercise has focussed on four EMEP intensive measurement periods (Aas *et al.*, 2012):

- 1 Jun - 30 Jun 2006
- 8 Jan - 4 Feb 2007
- 17 Sep - 15 Oct 2008
- 25 Feb - 26 Mar 2009

Differently to the previous inter-comparison exercises, most of models have been run in ED-III with the same input data (emissions, meteorology, boundary conditions) and over the same domain (domain extension and resolution) with some rare exceptions. Participating models were applied over four different periods, within a rather limited number of years thus allowing to evaluate the influence of different meteorological conditions on model performance. All models except RCG have run the four periods. The institutes/laboratories participating in the ED-III with their models are reported in Table 1. The other participants JRC, BSC, CIEMAT, CONCAWE, AERIS EUROPE, LMD/IPSL and University of Brescia contributed to the project bringing their expertise in air quality modelling, model evaluation and management of observational data. The ED-III framework was also used to assess the impact of the horizontal resolution on the performance of air quality models (Schaap *et al.*, 2015).

The ED-III exercise allows a very comprehensive inter-comparison and evaluation of chemistry transport models performance with a join analysis of some meteorological variables. A first evaluation on the 2009 campaign with an interim version of models was already published in Bessagnet *et al.* (2014). Moreover, the selected periods coincide with EMEP intensive measurement periods so that an extended set of observational data were available. Therefore, in addition to EMEP operational monitoring data, also size disaggregated (in $PM_{2.5}$ and $PM_{10}$) aerosols and hourly measurements for studying diurnal cycles have been employed. Additional AirBase data (Mol and de Leeuw, 2005) were used to evaluate the impact of meteorology on air pollutant concentrations. Finally, the exercise was performed under strict requirements (with some exceptions) concerning the input data. As a consequence, there were very limited differences in the models set up, representing a sort of sensitivity analysis to several aspects of the modelling chains. The objective of this paper is twofold, (i)





to present the exercise, the input data and the participating models, and (ii) to analyse the behaviour of models in the four campaigns focussing on the criteria pollutants $PM_{10}$, $PM_{2.5}$, $O_3$, $NO_2$ and $SO_2$ and relevant meteorological variables. Complementary analyses of depositions fluxes and PM composition data at high temporal resolution will be discussed in companion papers in order to better understand the behaviour of models.

5 ## 2    Description of models

### 2.1    Overall description of models

The models are synthetically described in Table 2 and Table 3. All the models were run on the same domain at 0.25°x0.25° resolution in longitude and latitude, shown in Fig. 1, except CMAQ. CMAQ simulations were performed on a Lambert-conformal conic projection with the standard parallels at 30 and 60 degrees and a grid of 112 by 106 cells of size 24km x 10    24km. The results of the CMAQ simulations were interpolated to the prescribed EURODELTA grid.

Participants delivered both air concentrations and meteorological parameters. Most of variables were delivered on hourly basis, while dry and wet deposition fluxes were provided on a daily basis. The output species include, among others: $O_3$, $NO_2$ and $SO_2$, total PM mass concentrations both in 2.5 and 10 μm fractions ($PM_{10}$ and $PM_{2.5}$), Secondary Inorganic 15    Aerosols (ammonium, sulphate and nitrate) and other PM components relevant for the analysis as well as wet deposition of sulphur and nitrogen compounds were also collected and will be used in companion papers. The delivered air concentrations should approximately correspond to the standard measurement height (typically 3 m) and were directly derived from the first layer, except for LOTOS-EUROS and EMEP that corrected the concentrations from the first layer to be representative of 3-m concentrations. The $PM_{2.5}$ and $PM_{10}$ concentration are calculated as follows in each model:

20 $$PM_{xx} = PPM + SO_4^{2-} + NO_3^- + NH_4^+ + Sea\ Salt + SOA + Dust$$

where PPM stands for Primary Particulate Matter and includes Elemental carbon, Primary organic aerosol and primary non-carbonaceous aerosol, SOA represents Secondary Organic Aerosol, Sea Salt and Dust represent the contribution of the corresponding natural processes.

The participating models differ in the availability of PM components and formation routes. For instance, EMEP, LOTOS-25    EUROS and RCG contain coarse mode nitrate formation, whereas the others do not. In CMAQ additional anthropogenic dust is calculated as 90% of unspecified PM coarse emissions and attributed to fugitive dust (Binkowsky and Roselle, 2003). CAMx did not activate the sea salts parameterisation in this exercise.

Based on the set-up of models and completeness of datasets, an "ENSEMBLE" called **ENS** has been built based on mean values of model outputs. To compare the behaviour of models for all pollutants and campaigns, only CHIMERE, MINNI, 30    LOTOS-EUROS and EMEP constitute the "ENSEMBLE". CAMx, CMAQ and RCG were not included in the ensemble for three reasons: (i) CAMx did not account for sea salts giving rise to very different PM patterns over the oceans and seas, (ii) CMAQ used a different meteorology and (iii) RCG did not cover the four campaigns.



## 2.2    PBL height and mixing in models

### CAMx

In ED-III the Planetary Boundary Layer was directly taken from the IFS-ECMWF data (Integrated Forecast System of the
European Centre for Medium-Range Weather Forecasts). The PBL height was then used by CAMx pre-processor to derive
$K_z$ profiles. For ED-III the O'Brien scheme (1970) has been used to derive $K_z$ profiles as Eq.1:

$$K_z = K_A + \frac{(z-z_A)^2}{(z_A-z_B)^2}\left\{K_B - K_A + (z-z_B)\left(K_B' + 2\frac{K_B-K_A}{z_A-z_B}\right)\right\} \tag{Eq. 1}$$

Where $K_z$ is a value of $K_A$ at the height of the atmospheric boundary layer, $z_A$, and $K_B$ at the height of the surface layer $z_B$, the
so-called constant-flux layer. Minimum $K_z$ values have been set to 1. Any values of $K_z$ calculated below, will be set to this
value. By default, CAMx employs a standard "K-theory" approach for vertical diffusion to account for sub-grid scale mixing
layer-to-layer.

### CHIMERE

In this study, the Planetary Boundary Layer is directly taken from the IFS ECMWF data. Horizontal turbulent fluxes were
not considered. Vertical turbulent mixing takes place only in the boundary layer. The formulation uses K-diffusion following
the parameterization of (Troen and Mahrt, 1986), without counter-gradient term. In each model column, diffusivity $K_z$ is
calculated as Eq. 2:

$$K_z = k\,w_s\,z\left(1 - \frac{z}{h}\right)^{1/3} \tag{Eq. 2}$$

where $w_s$ is a vertical velocity scale given by similarity formulae.

-    In the stable case (surface sensible heat flux < 0): $w_s = u_*/(1 + 4.7\,z/L)$
-    In the unstable case: $w_s = (u_*^3 + 2.8ew_*^3)^{1/3}$

where $e = max(0.1,z/h)$, $L$ is the Monin-Obukhov Length, $w_*$ is the convective velocity scale, $u_*$ the friction velocity and $h$
the boundary layer height. A minimal $K_z$ is assumed, with a value of 0.01 m$^2$ s$^{-1}$.

$K_z$ and the wind speed were corrected in urban zones according Terrenoire *et al.* (2015) by applying a correction factor to
limit the diffusion within the urban canopy, but this correction has very little effect at this resolution.

### CMAQ

The boundary layer height in COSMO is calculated with the turbulent kinetic energy (TKE) method (Doms *et al.* 2011).
CMAQ directly used the PBL fields from COSMO.

In CMAQ the vertical turbulent mixing is estimated using the Asymmetric Convective Model scheme version 2 (ACM2,
Pleim, 2007a,b). The ACM2 replaces the simple eddy viscosity (K-theory) scheme. ACM2 scheme allows the non-local
mixing, which means upward turbulent mixing from the surface across non-adjacent layers through the convective boundary





layer. Pleim (2006) compared the eddy viscosity and the ACM2 schemes in CMAQ, finding that the ACM2 schemes tends to predict larger concentrations of secondary pollutants and smaller concentrations of primary pollutants at the surface, and has a more well-mixed profile in the PBL than the eddy viscosity scheme.

CMAQv5 has also an improved version of the minimum allowable vertical eddy diffusivity scheme. The new version interpolates between urban and nonurban land cover, allowing a larger minimum vertical diffusivity value for grid cells that are primarily urban. Moreover, the minimum eddy diffusivity were reduced from 0.5 m$^2$ s$^{-1}$ to 0.01 m$^2$ s$^{-1}$, and from 2.0 m$^2$ s$^{-1}$ to 1.0 m$^2$ s$^{-1}$ for urban areas.

**EMEP**

The mixing height is calculated using a slightly modified Richardson number ($Ri_B$) following Jeričevič *et al.* (2010) and defined as the lowest height at which the $Ri_B > 0.25$. Finally, the PBL is smoothed with a second order Shapiro filter in space. The PBL height is not allowed to be less than 100 m or exceed 3000 m.

The initial calculation of the vertical exchange coefficients is done using the Ri number and wind speed vertical gradient for the whole domain. Then, $K_z$ values within the PBL are recalculated based on Jeričevič *et al.* (2010) for stable and neutral conditions. For unstable situations $K_z$ is calculated based on the similarity theory of Monin-Obukhov for the surface layer, whereas $K_z$ profiles from O'Brian (1970) are used for the PBL above the surface layer. For more detail see Simpson *et al.* (2012).

**LOTOS-EUROS**

The first model layer is by definition the mixing layer, with height equal to the boundary layer height as given by ECMWF. Horizontal diffusion is not used, but for vertical mixing the vertical diffusion coefficient is calculated according to Eq. 3.

$$K_z = \frac{\kappa u^*}{\Phi(z/L)}$$
(Eq. 3)

where $\kappa$ is the von Karman constant, $u^*$ the friction velocity, $\Phi$ the functions proposed by Businger (1971) for stable, neutral or unstable atmosphere, z the height and L the Monin-Obhukov length. The friction velocity is calculated depending on the wind at reference height (10 m), the Businger functions and the roughness length per land use class. The vertical structure of LOTOS-EUROS is determined by the mixing layer height, with a shallow surface layer (25 m) to avoid too fast mixing of near-surface emissions and a second layer equal to the mixing layer as given by ECMWF.

**MINNI**

Friction velocity and Monin-Obukhov length are determined by using:-Holtslag and van Ulden (1983) iterative scheme for unstable conditions and Venkatram (1980) iterative method for stable conditions. Micro-meteorological parameters over





water are derived with the profile method, using air-sea temperature difference (Hanna *et al.*, 1985), with the needed roughness length, depending on wind speed, supplied by the Hosker (1974) parameterization.

During daytime both convective and mechanical heights are determined, keeping then the maximum value between the two parameters. The convective height is calculated following the Maul (1980) version of Carson (1973) algorithm, essentially based on heat conservation equation; mechanical mixing height is instead estimated by using Venkatram (1980) algorithm. During nighttime, the Bulk Richardson number method is applied (Sorensen, 1998), in which the height of the boundary layer is given by the smallest height at which the bulk Richardson number reaches the critical value fixed to 0.25.

**RCG**

The mixing layer depth in the model is the height of the layer closest to the input boundary layer height taken from the IFS ECMWF data. Vertical diffusion parameters for stable and unstable conditions are derived using the Monin-Obukhov similarity theory for the description of the structure of the diabatic surface layer. The friction velocity and Monin-Obukhov-length are calculated iteratively depending on the 10 m wind, the stability corrections factors and the land use dependent on roughness length.

## 3 Input data

### 3.1 Anthropogenic emissions

The first step in the emission preparation was to calculate the spatial pattern of emissions for the reference year 2007, that was selected because it was a key year for the TNO-MACC inventory (Kuenen *et al.*, 2011). The anthropogenic emission input was harmonized following the methodology described in Terrenoire *et al.* (2015). The total emissions per sector and country were then scaled to the year corresponding to the campaigns: 2006, 2007, 2008 and 2009. Emission categories are numbered in 11 classes called SNAP (Selected Nomenclature for Air Pollutants): (1) Public Power stations, (2) Residential and Comm./inst. Combustion, (3) Industrial combustion, (4) Production processes, (5) Extraction and distribution fossil fuel, (6) Solvents use, (7) Road traffic, (8) Other mobile sources (trains, shipping, aircrafts, ...), (9) Waste treatment, (10) Agriculture, (11) Natural (models used their own natural emissions in this exercise).

The gridded distribution of anthropogenic emissions was provided by INERIS and it was based on a merging of different databases from:

- TNO $0.125° \times 0.0625°$ emissions for 2007 from MACC (Kuenen *et al.*, 2011)

- EMEP $0.5° \times 0.5°$ emission inventory for 2009 (Vestreng *et al.*, 2007)

- Emission data from the GAINS database (http://gains.iiasa.ac.at/gains).



Emission re-gridding was based on INERIS expertise and performed by means of various proxies:

- population data coming from the EEA database merged with global data (from the Socioeconomic Data and Applications Center http://sedac.ciesin.columbia.edu) to fill gaps in Europe.;
- the US Geophysical Survey landuse at 1 km resolution (http://www.usgs.gov/).
- French bottom-up emission data for wood combustion to derive a proxy;
- EPER data for industries; the EPER Decision is based on Article 15(3) of Council Directive 96/61/EC (EC, 1996) concerning integrated pollution prevention and control. EPER is a web-based register, which enables the public to view data on emissions to water and air of 50 key pollutants from large and medium-sized industrial point sources in the European Union.

The TNO-MACC dataset provides two distinct datasets (i) large point sources (LPS) with the coordinates of stacks and (ii) surface emissions on a fine grid (0.125°×0.0625°). In the gridding process, the first step consisted in summing up LPS emissions from the TNO-MACC emissions inventory for 2007 with surface emissions to obtain total emissions as in the EMEP inventory. LPS were aggregated with surface emissions because no data were available to calculate plume rise heights for point sources emissions. For the various SNAP, the processing steps were the following:

- SNAP 2: The country emissions were re-gridded with coefficients based on population density and French bottom-up data, the methodology (Terrenoire *et al.*, 2015) was extrapolated to the whole Europe. For $PM_{2.5}$ emissions, the annual EMEP national totals were kept except for the countries: Czech Republic, Bosnia and Herzegovina, Belgium, Belarus, Spain, France, Croatia, Ireland, Lithuania, Luxemburg, Moldavia, Republic of Macedonia, Netherland, Turkey. For these countries, $PM_{2.5}$ emissions from GAINS were used. Additional factors were applied on two Polish regions (×4 or ×8) for $PM_{2.5}$ and $PM_{10}$ emissions (Kiesewetter *et al.*, 2015). The former activity in coal mine regions still leads to high emissions of PM due to domestic uses of coal.

- SNAP 3,7,8,9,10: TNO-MACC emission spatial distribution was used as proxy to regrid EMEP 0.5°x0.5° annual totals into the finer modelling grid.

- SNAP 1,4,5,6: EMEP 0.5°x0.5° emissions were regridded by using "artificial area", except for industries where EPER data were used.

For countries where TNO-MACC emissions were not available, the EMEP 0.5°×0.5° emissions were used (Iceland, Liechtenstein, Malta and Asian countries) and regridded with adequate proxies ("artificial landuse", EPER data for industries).

The following emitted species were used in the models: methane (this species comes from the TNO-MACC inventory), carbon monoxide, ammonia, sulphur oxides, non methane volatile organic compounds (NMVOC), nitrogen oxides, primary particulate matter.

Residential emissions of particulate matter are dominant in wintertime in most of countries; they come from wood burning or coal uses. Germany, Sweden, Spain clearly have the lowest levels of emissions. Romania, Poland and France have the highest levels of emissions (Terrenoire *et al.*, 2015).

The time profiles are those used in Thunis *et al.* (2008). Three types of profiles were provided:

- Seasonal factors : one value per species, month, activity sector and country
- Weekly factors : one value per species, day type (Monday – Sunday), activity sector and country





- Hourly factors : one value per hour (local time), species and activity sector

The vertical injection profile in CTMs was prescribed according to Bieser *et al.* (2011) where industrial sectors and residential heating were assigned in lower levels compared to the usual default profiles (Mailler *et al.*, 2013).

Since only $PM_{2.5}$ and coarse PM emissions were provided by the EMEP, a PM speciation profile provided by IIASA (Personal Communication from IIASA) was used to estimate the fraction of Non-carbonaceous species, Elemental Carbon and Organic Matter per activity sectors and country. Models used their own split for NOx, SOx and NMVOC emissions. This emission inventories did not account to recent changes in the way to account for Semi Volatile Organic Compounds from wood burning emissions as discussed in Denier van der Gon *et al.* (2015).

## 3.2    Natural emissions

*Biogenic VOC emissions from vegetation*

CHIMERE and MINNI used the version 2.04 of the MEGAN model while CAMx uses the 2.1 version (Guenther *et al.*, 2006, 2012). The Model of Emissions of Gases and Aerosols from Nature (MEGAN) is a modelling framework for estimating fluxes of biogenic compounds between terrestrial ecosystems and the atmosphere using simple mechanistic algorithms to account for the major known processes controlling biogenic emissions. It is available as an offline code and has also been coupled into land surface and atmospheric chemistry models.

EMEP, LOTOS-EUROS and RCG used parameterizations derived from Simpson *et al.* (1999) for the temporal variations according temperature and light, with maps of tree species from Koeble and Seufert (2001).

CMAQ used the BEIS (Biogenic Emission Inventory System: Vukovich and Pierce, 2002) module developed by the US EPA. BEIS estimates volatile organic compound (VOC) emissions from vegetation and nitric oxide (NO) and carbon monoxide (CO) emissions from soils. Because of resource limitations, recent BEIS development has been incorporated into the Sparse Matrix Operational Kernel Emissions (SMOKE) system, so that the native version of BEIS is built within the SMOKE architecture.

*Soil NO emissions*

CHIMERE, CAMx and MINNI used the version 2.04 of the MEGAN model to calculate the NO emissions. RCG used a parameterization of NO emissions described in Simpson *et al.* (1999). LOTOS-EUROS did not include NO emissions in this simulation. CMAQ used the BEIS (Biogenic Emission Inventory System) module developed by the US EPA. NO soil emissions for EMEP are described in Simpson *et al.* (2012)

*Sea salt emissions*

All models host very different schemes based on Monahan (1986) for CHIMERE and updates from Martensson *et al.* (2003) for LOTOS-EUROS, and Gong *et al.* (1997) for RCG. CMAQ and MINNI used the Zhang *et al.* (2005) parameterization



and CAMx had no sea salts for this exercise. EMEP used parameterisation from Monahan (1986) for larger sizes of sea spray and Martensson *et al.* (2003) for smaller sizes.

CMAQ emits also sea salts sulphate using a fraction of 7.76% of emitted sea salts split into the accumulation and coarse modes.

*NO emissions from lightning*

Climatologies of NO emissions from lightning is based on Köhler *et al.* (1995) in EMEP. The other models do not account for this kind of emissions.

*Wildfire emissions*

Fire emissions were provided by the GFASv1.0 database (Kaiser *et al.*, 2012) only for the 2006 campaign. The Global Fire Assimilation System (GFASv1.0) calculates biomass burning emissions by assimilating Fire Radiative Power (FRP) observations from the MODIS instruments onboard the Terra and Aqua satellites. It corrects for gaps in the observations, which are mostly due to cloud cover, and filters spurious FRP observations of volcanoes, gas flares and other industrial

activity. For all models the wildfire emissions were assigned in the whole PBL layer.

*Dust emissions*

For CAMx, CHIMERE and CMAQ, no dust module is activated for this exercise. For these three models, natural dust only comes from the boundary conditions. For EMEP, windblown dust parameterisation is documented in Simpson *et al.* (2012),

road dust calculations are included in the calculations from Denier van der Gon *et al.* (2009). LOTOS-EUROS contains emission parameterizations for several sources of mineral dust (Schaap *et al* 2009). Only wind-blown dust, resulting from wind erosion of bare soil, was taken into account here, together with dust from boundary conditions. Other sources (agricultural activities, road dust resuspension) were not activated in ED-III. In MINNI, dust emissions from local erosion and particle resuspension (Vautard *et al.*, 2005) with attenuation in the presence of vegetation from Zender *et al.* (2003) is

activated in this exercise. RCG considers resuspension of mineral aerosol as a function of friction velocity and the nature of soil. Two mechanisms are treated: direct release of small dust particles by the wind (Loosmore and Hunt, 2000), and indirect release by collisions with bigger soil grains, that are lifted by the wind but return to the surface because of their weight (saltation process from Claiborn *et al.*, 1998).

**3.3   Meteorology**

All models except CMAQ and RCG shared the same meteorological dataset at 0.2° resolution based on ECMWF IFS (Integrated Forecast System) calculations.



Because of its importance for applications (*e.g.* in air pollution modelling), the boundary layer height was diagnosed in ECMWF was made available. The parameterization of the mixed layer (and entrainment) uses a boundary layer height from an entraining parcel model. But in order to get a continuous field, also in neutral and stable situations the bulk Richardson method proposed by Troen and Mahrt (1986) is used as a diagnostic, independent of the turbulence parameterization.

Boundary layer height is defined as the level where the bulk Richardson number, based on the difference between quantities of energy at that level and the lowest model level, reaches the critical value $Ri_{cr} = 0.25$.

For RCG, a different meteorological data set was used. The 3D-data for wind, temperature, humidity and density were produced employing a diagnostic meteorological analysis system developed at Freie Universität Berlin and based on an

optimum interpolation procedure on isentropic surfaces. The system takes into account all available observed synoptic surface and upper air data as well as topographical and land use information (Reimer and Scherer, 1992). Rain data, cloud data and boundary layer heights were retrieved from the IFS data set. Boundary layer parameters as friction velocity and Monin-Obukhov-lenghth were calculated on-the-fly applying standard boundary layer theory.

The CMAQ model used meteorological variables calculated with the COSMO model in CLimate Mode (COSMO-CLM) version 4.8 clm 11. The COSMO model is the non-hydrostatic operational weather prediction model applied and further developed by the national weather services joined in the COnsortium for SMall scale MOdeling (COSMO) described in Bettems *et al.*, (2015).

### 3.4    Boundary conditions

In this study, the MACC reanalysis were used as input data for the boundary conditions (Inness *et al.*, 2013; Benedetti *et al.*, 2009). The MACC II project (Modelling Atmospheric Composition and Climate) is establishing the core global and regional atmospheric environmental service delivered as a component of the COPERNICUS initiative. The reanalysis production stream provides analyses and 1-day forecasts of global fields of $O_3$, CO, $NO_2$, $SO_2$, HCHO, $CO_2$, $CH_4$, and aerosols. Other

reactive gases are available from the coupled chemistry transport model. The reanalysis cover the period $2003 - 2011$ with a one month spin-up. It runs at approximately 78 km by 78 km horizontal resolution over 60 levels. The coupled chemistry transport model has the same 60 vertical levels and a horizontal resolution of 1.125 degrees x 1.125 degrees. For aerosols only elemental carbon, organic carbon, dust and sulphate were used.

Stratospheric ozone fields from the MACC reanalysis agree with ozone sondes and ACE-FTS data (Atmospheric Chemistry

Experiment Fourier Transform Spectrometer) within ±10% in most seasons and regions. In the troposphere the reanalysis shows biases of −5% to +10% with respect to ozone sondes and aircraft data in the extratropics, while larger negative biases are shown in the tropics. Area-averaged total column ozone agrees with ozone fields from a multi-sensor reanalysis data set within a few percent. For aerosols, the observed Aerosol Optical Depth (AOD) is assimilated in the model with a feedback



on individual PM species (sea salts, dust, elemental carbon, organic carbon and sulphate). When available, the MACC reanalysis is compared with observations, the model acronym in the supporting material is MACCA.

## 4 Observation dataset and statistics

### 4.1 Air pollutant concentrations

The evaluation was carried out on the available EMEP standard monitoring (Tørseth *et al.*, 2012) and intensive period observations for 2006, 2007, 2008 and 2009 (Aas *et al.*, 2012) on hourly and daily bases (see supplementary material S8 for the description of background sites). Elevated sites above 1500 m in altitude have been excluded from the analysis. The measurements were downloaded from the EBAS database. (http://ebas.nilu.no/). Additional AirBase data (Mol and de Leeuw, 2005) were used to evaluate the impact of meteorology on air pollutant concentrations in section 7.2.

It is important to note that daily measurements for a day N is the averaged value between day N HH:00 and day N+1 HH:00, with HH usually varying in the range [00, 09]. For most of species, measurements on daily and hourly bases are not necessarily performed for the same set of stations. Deposition and the PM composition are also available; the dataset will be detailed in the companion papers.

### 4.2 Meteorology

**Temperature and wind speed**

The temperature, wind speed and precipitation measurements come from 2016 synoptic stations in Europe reported by the European meteorological centres. The data are provided on an hourly basis. The temperature is measured at 2 m and the wind speed at 10 m. Some meteorological data are also reported at some EMEP. At EMEP sites daily accumulated measurements (e.g. precipitation) for a day N represent the integral between day N HH:00 to day N+1 HH:00, with HH usually varying in the range [00, 09].

**Planetary Boundary Layer (PBL) height**

The soundings data were extracted from the University of Wyoming database. For each site and for each day, two soundings are available at 00:00 and 12:00. The provided meteorological parameters are: pressure (hPa), the corresponding height above ground level (m), dew point temperature (°C), relative humidity (%), mixing ratio (g kg$^{-1}$), wind direction (degrees) and wind speed (expressed in knot and converted in m/s by applying the conversion factor 0.514), potential and virtual potential temperature (K). For the present study, data were extracted over 77 stations in Europe. The boundary layer height is estimated using the calculation of the Bulk Richardson number profile and searching for the altitude where the critical value of $Ri_{cr}=0.25$ is reached. The analysis was limited to the first 25 vertical points, roughly corresponding to an altitude of



5000m above ground level. Being the boundary layer height a concept valid only for convective periods, only the sounding of 12Z were analyzed and used for the models evaluation.

In addition to the previous PBL data, hourly heights of the atmospheric boundary layer were calculated from LIDAR measurements in a background site near Paris (SIRTA in Palaiseau). A new objective method for the determination of the atmospheric boundary layer depths using routine LIDAR measurements have been used (Pal *et al.*, 2013).

### 4.3    Error statistics for the evaluation of model performances

The errors statistics considered in this report are presented in Table 4. In supplementary material S0-S1 the statistic performances of all models for the four campaigns are reported. For a given pollutant or meteorological variable, model performance is computed for a common set of stations (over the same common geographic area). All maps of pollutant concentrations and meteorological variables concerning individual models and ensemble are provided in supplementary material (S2-S6).

For the analysis of the "ensemble" a coefficient of variation *VAR* is defined as follows in Eq. 4:

$$VAR = \frac{1}{C_{ENS}} \sqrt{\frac{1}{M} \sum_m (C_m - C_{ENS})^2} \qquad \text{(Eq. 4)}$$

With $C_m$ the concentration of individual model $m$ included in the *ensemble* (CHIMERE, LOTOS-EUROS, MINNI and EMEP), M is the number of models, and $C_{ENS}$ is the *ensemble* mean concentration.

### 5    Evaluation of the meteorology

Some general feature for each campaign can be provided, they are issued from the NOAA (National Oceanic and Atmospheric Administration) global analysis (https://www.ncdc.noaa.gov/sotc/global/).

June 2006 temperatures were above average everywhere in Europe with low precipitation except in Balkan countries and Spain compare to the 1961-1990 base period.

January 2007 is characterized by windy conditions in Europe with cool temperature above average everywhere in Europe except in Spain where temperatures were close to the average values. In the beginning of February temperature decreased in Scandinavia. Precipitation were low over the Mediterranean basin but above the climatic average compare to 1961-1990 base period in the rest of Europe.

In September- October 2008, no strong general characteristics were recorded; this transition period was characterized by slight negative temperature anomalies in the western part of Europe, mainly France, United Kingdom and north of Spain.

After some cold spells end of February, March 2009 turned cooler with on average warmer temperatures compare to the 1961-1990 base period. Precipitation anomalies were negative in the west part of Europe and positive in the central and east part of Europe.

**Temperature**



As summarized in supplementary material S0, the models using ECMWF data show comparable high temporal correlation coefficients based on hourly values over the whole domain (0.88 < R < 0.94), with highest correlations values in northern Germany and France when looking on a daily basis. Correlations are lower whatever the model over north of Italy and Austria. On average for the considered period, the bias is negative for all models in the range [-0.3 K, -0.7 K] for CAMx,

CHIMERE, EMEP and LOTOS-EUROS. The negative bias for this group of models is more important for the two wintertime campaigns, however in Switzerland and Austria this bias exceed -2K whatever the campaign. Since this group of models shares the same meteorology, the error statistics are very similar; the discrepancies are due to the different interpolation methods from ECMWF data to the CTM grid.

RCG displays a very low absolute bias close to zero for the 2009 campaign, and CMAQ displays the lowest negative bias up

to -2K for the 2009 campaign. CMAQ has lower correlation coefficient particularly in Germany and Poland for the 2008 and 2009 campaigns.

As displayed in Fig. 2, the negative bias is driven by afternoon temperatures that are underestimated by all models, this statement is valid for all campaigns. The nighttime temperatures are more in line with the observations. The RCG diurnal cycle is rather different with a flatter profile but for the other models using ECMWF or COSMO data, the general pattern is

well captured.

**Wind speed**

All the models using ECMWF data overestimate the wind speed from +0.1 to +0.9 m s$^{-1}$, while CMAQ, driven by COSMO, showed on average the lowest absolute bias. The biases are the highest for the two winter (2009) and fall (2008) campaigns,

while for the summer campaign (2006) the biases are lower. It is worth noting that the 2007 campaign was the most windy period, showing a mean observed wind speed of 4.77 (m/s).

Bias is generally higher in eastern and northern Europe than in western and Mediterranean areas. In Europe the spatial pattern of biases shows high positive bias in several coastal areas and negative bias in mountainous areas (Alps). This clearly points out a problem in some regions for the calculation of some emissions directly relying on IFS U10 fields. According to

Ingleby *et al.* (2013) ECMWF 10 m wind speeds are slightly overestimated especially at night. In the IFS only 10m winds are used from ships over the oceans for data assimilation (problem of station representativeness for inland stations). Moreover, errors on wind speed measurements are stronger for low winds. For the lowest winds generally observed during nightime the comparison of the predicted diurnal cycle with observations show a largest positive bias at night than during the afternoon (Fig. 2), this behaviour could lead to an overestimation of the advection process.

Time correlations are better for models using ECMWF data but all models exhibit low correlations over the Alps regions (North of Italy, South East of France, Switzerland and Austria). The RCG model shows higher correlation coefficients over northern Europe (Finland and Sweden) for the 2009 campaign.

**Planetary boundary layer (PBL) and mixing**





As explained in section 4.2, the observed PBL height was calculated at 12:00 because of methodology hypotheses, except at the SIRTA site where hourly measurements are available for 2008 and 2009. All models have a negative bias, the lowest RMSE are displayed for CAMx and CHIMERE which use the ECMWF PBL, the biases are in the range -237 m and -100 m for these two models. It is worth noting that CAMx and CHIMERE exhibits exactly the same performance, while LOTOS-

EUROS and EMEP that should adopt IFS PBL too, show partially different performance, suggesting that the latter models partially recomputed boundary layer height. The largest underestimation of the PBL height is usually found for MINNI particularly for the 2006 campaign (up to -616 m) and EMEP (up to -451 m) and the correlation coefficients for these models are lower compared to the others. CMAQ has the lowest bias for most of campaigns. The temporal correlations displayed in supplementary material S0 are the best for models using the IFS PBL, the main discrepancies are observed for the 2006

campaign with several sites in Europe with negative correlations. The largest negative biases are observed in the south of the domain, in these regions CMAQ performs better. In some regions over the Mediterranean basin, particularly in coastal areas, the MINNI's PBL is sometimes strongly biased up to -1000m. The obtained results suggest that either the Carlson algorithm or the micro-meteorological parameterization implemented by MINNI tends to underestimate the intensity of convention.

The spatial representation of the PBL for the 2009 campaign shows higher differences between the models mainly over the ocean and seas where the coefficient of variation reaches 40% in some areas (Fig. 3). While LOTOS-EUROS, CHIMERE, RCG and CAMx use the PBL from ECMWF PBL with some differences on spatial and time interpolations, the other models use their own parameterizations discussed in section 2.2. The diurnal cycles displayed in Fig. 3 show that MINNI simulate higher PBL at night and lower PBL during daytime compare to ECMWF, the differences of the afternoon PBL is quite

important over countries influenced by the ocean like the Great Britain. CMAQ and EMEP simulate over France and Great Britain the highest PBL at night. The hourly times series at the SIRTA site confirm the underestimation of the ECMWF PBL but at this station the negative bias of MINNI has the same order of magnitude as the other models. The correlations based on hourly values are still lower for CMAQ, EMEP, MINNI (below 0.50) compared to the models using ECMWF data.

The differences of treatments of the advection and mixing as reported in section 2.2 lead to differences of the dispersion. Fig.

4 shows the mean coefficient of variation of CO concentrations predicted by the model sharing the same raw meteorology (IFS) for the 2006 campaign. This pollutant can be considered as a tracer with low influences of deposition and chemistry processes, most of the differences on concentrations are related to transport and mixing. The figure clearly shows that mixing on emissions areas, such as big cities, produces the highest differences exceeding 20% of variations. Besides of urban areas, the highest coefficients of variation are observed over the seas and ocean that are related to the differences of PBL predicted

by the models (Fig. 3), elsewhere this coefficient remains below 10%.



# 6    Overall model performance evaluation on criteria pollutants

## 6.1    Ozone

The models performances (supplementary material S1) are very different from campaign to campaign. Most of models overestimate ozone concentrations in 2006, 2007 and 2008 (Fig. 5). Only the 2009 campaign show a systematic
underestimation of observed ozone concentrations from -5 to -16 µg m$^{-3}$. The large positive bias in 2007 and negative in 2009 are largely explained by the boundary conditions that are biased respectively of +8 and -20 µg m$^{-3}$ (Supplementary material S1). Correlations are similar for all models in the range 0.5-0.6, only CMAQ has lower correlations on average. For the summertime campaign 2006 CHIMERE and CMAQ display the lowest correlation for daily averaged concentrations but CHIMERE has the lowest bias with EMEP. For this campaign most models underestimate concentrations in mountainous
regions in Spain and over the Alps (Fig. 6). The models tend to over predict ozone concentrations on background stations influenced by large urban areas like GR01 station in Greece and IT01 close to Roma. All models simulate high ozone concentrations over the Mediterranean sea, most of them behaves satisfactorily in Malta and Cyprus stations confirming the ozone concentrations pattern over the seas for the "ensemble" shown in Fig. 6. The diurnal cycles in Fig. 7 reflect the overall performances depicted previously. All models fairly simulate the timing of the daily peak. For campaign 2007, except
MINNI the models overshoot during nighttime and daytime. For campaign 2008, the very good shape of the LOTOS-EUROS diurnal cycle is remarkable. For the summertime campaign 2006, CHIMERE and EMEP provide on average the best diurnal cycles. Focussing on 2006 and 2008 campaigns, the two campaigns which are not biased by the boundary conditions, LOTOS-EUROS show the best performances regarding the bias. For these two campaigns, CAMx has a strong positive bias particularly at night. CAMx shares exactly the same PBL height of CHIMERE, but night-time performance of
the two models are rather different. This result confirms that during stable conditions the pollutant concentration is influenced not only by the PBL height, but also by the overall reconstruction of vertical dispersion.
In Fig. 6, the right side is the gridded coefficient of variation that is a standardized measure of the dispersion of model results. It is defined as the ratio of the standard deviation to the mean of models. This coefficient is very low for the 2006 campaign, below 10%, the models have different responses along the ship tracks. The coefficients of variation are the
highest for the 2007 campaign (supplementary material S2) associated with low performances of the "ensemble" (high normalized root mean square errors). Not only the bias is affected by global boundary conditions, but also this result indicates that biased ozone boundary conditions globally impair the normalized statistics confirming the non linearity of ozone chemistry. France, Spain and Norway show the lowest coefficient of variation indicating a more coherent behaviour among the models, but not necessarily corresponding to better model performance than other areas.
At Mace Head (IE31) located on the west part of the domain the time series of model results versus ozone observations show flat shape for the two winter campaigns with very low time correlations in 2009 (Fig. 8). The best correlation coefficients are observed for 2006 and 2008, the models are able to capture the peaks. At this station the negative bias mentioned in 2009 is roughly the same for LOTOS-EUROS, MINNI and RCG and comparable to the MACC analysis (-20 µg m$^{-3}$), the other





models CAMx, EMEP, CHIMERE and CMAQ have a lower absolute bias (about -10 µg m$^{-3}$). This behaviour shows that boundary conditions are quickly modified certainly because the regional models restoring their own chemical equilibrium in relation with dynamical processes like deposition and vertical dispersion.

## 6.2   Nitrogen dioxide

For NO$_2$, all models perform similarly in terms of correlation with value in the range 0.6-0.7 (Fig. 5 and supplementary material S1). The spatial correlation is much higher in the range 0.7-0.9 for all models. Only CMAQ strongly overestimates the mean concentrations and CAMx underestimates the concentrations for all campaigns. This underestimation of NO$_2$ concentrations is certainly related to rather high ozone concentrations.

The spatial patterns of the "ensemble" shown for 2009 (Fig. 9) display high concentrations over the Benelux, North Italy, the biggest cities and over the shipping tracks. The bias of the "ensemble" is rather good except for one station in Serbia (RS05) with high observed values, probably due relevant local sources. The gridded coefficients of variation provided in Fig. 9 show that most of differences between models are observed over remote areas for from emission regions even if errors are expected to occur more frequently for low values. As shown for a non reactive species like CO, the mixing of close to

emissions is responsible for model output differences, this effect can be clearly seen over the East Mediterranean for maritime emissions where the PBL is different from model to model. Over lands the NO$_2$ chemistry and the different biogenic NO emissions explain a large part of the differences far from urban areas. As shown in Fig. 9, the root mean square errors of the models are the highest for the stations close to the emission areas. The diurnal cycles in Fig. 10 show a general underestimation during the afternoon. It should be pointed out that the observed NO$_2$ concentrations can be slightly

overestimated because of sampling artefact (evaporation of nitric acid). In the observations, the presence of two peaks on NO$_2$ concentrations is related to the traffic emissions peaks occurring in the morning and the evening. The timing of the peak occurrences is also modulated by the meteorology, for the 2006 and 2008 campaigns performed with identical summer time shift we clearly see a time shift of +1 and -1 respectively for the morning and evening peaks corresponding to a later rise and earlier fall of the PBL. Thus, as expected, the narrowest time lag between the two peaks is observed for the 2007 campaign.

Most of the models predict the first peak too early, particularly CHIMERE and CMAQ for the 2006 campaign, and the second peak generally occurs too late.

CMAQ shows the strongest night-time bias, that is the cause of the overall overestimation shown by the model in all campaigns. CMAQ was driven by a different meteorology that was characterized by very good performance with respect to both wind speed and PBL height mean bias. Conversely IFS-driven models overestimated night-time wind speed. As night-

time vertical mixing is mainly driven by mechanical forces, the obtained result suggests that models tend to underestimate mixing during stable conditions and, as a consequence, that IFS-driven models show better results for the wrong reason. Differently, differences in diurnal temperature between CMAQ and other models seem less relevant with respect to pollutant concentration.



### 6.3 Sulphur dioxide

The correlations are rather low for all models in the range 0.2-0.4 for the 2006 campaign to 0.5-0.6 for the 2007 campaign (Fig. 5 and supplementary material S1 for all statistics). Two groups of models are identified CAMx, MINNI and RCG that largely overestimate the concentrations and CHIMERE, CMAQ, EMEP and LOTOS-EUROS which are closer to the observations on average with the best performances on the RMSE. The overestimation of the first group of models could be explained as follows for MINNI which has the lowest PBL and RCG having the lowest wind speed. For CAMx, it is not explainable at this stage, an in-deep analysis with deposition and chemistry is necessary to understand this behaviour, this will be done in a companion paper. This involves a positive bias of the "ensemble" as shown in Fig. 11 (supplementary material S4) particularly in Western Europe; the normalized RMSE is frequently above 100% in most part of Europe. The main hot spots are located in the Eastern Europe in addition with high concentrations along the shipping routes. The coefficient of variation is the lowest over emission areas but very high in remote areas like over the oceans far from shipping tracks and over mountain areas. This is a first indication of the very different way to simulate the $SO_2$ chemistry and deposition processes in the models.

The diurnal cycles presented in Fig. 12 show a peak at 10:00 – 12:00. This peak is coherent with the hourly emission profiles of the industrial sector showing an emission peak at the same hour; however, most of models predict a larger decrease in the afternoon. Only CMAQ for the 2007 campaign captures satisfactorily the diurnal profile.

### 6.4 PM$_{10}$

Looking at the RMSE, on average the performances of the models are similar except CMAQ which has the highest values driven by low correlations and high negative biases for at least three campaigns (Fig. 5). All models underestimate the concentrations generally in the range -3 to -10 µg m$^{-3}$. Except CMAQ the correlations are in the range 0.4 – 0.6, but CHIMERE and EMEP reach 0.7 for the 2006 campaign. MINNI has the lowest absolute biases for the 2007, 2008 and 2009 campaigns. The "ensemble" provides a good picture of the PM$_{10}$ concentrations in Europe (Fig. 13 and supplementary material S5) except for two stations IT01 in Italy and CY02 in Cyprus with high recorded values. For CY02, high PM$_{10}$ concentrations are linked to high calcium concentrations due to dust events issued from North Africa. This dust event can be clearly observed for EMEP in Fig. 15. The spatial patterns show low concentrations below 5 µg m$^{-3}$ in remote Scandinavia and three hot spots in the Po valley, Benelux and South Poland. The coefficient of variations of model results is rather high over areas influenced by biogenic emissions as in Scandinavia, over the seas and arid areas. This coefficient is generally the lowest over the Western Europe. The best RMSE of the "ensemble" are observed for the summer campaign 2006 with values below 50% of the observations data.





EMEP has higher concentrations over the North Africa because the model generates dust in this part of the domain and sea salt concentrations are generally higher of the seas. EMEP and CHIMERE perform well for the spatial correlations (Table 5), EMEP captures better the high concentrations in the south of the domains whereas CHIMERE performs better over the Benelux (supplementary material S5). In 2008, RCG has particularly good spatial correlation compared to the other models.

The missing sea salt emission for CAMx is clearly observed over the ocean with very low $PM_{10}$ concentrations impairing the spatial correlations.

As shown in supplementary material S5, for the highest concentrations observed in 2008 and 2009, most of models underestimate the $PM_{10}$ by a factor of 2. For the 10% highest $PM_{10}$ concentrations, MINNI has the lowest underestimations for these two campaigns whereas EMEP behaves rather well for the 2006 campaign regarding the bias and the correlation.

As shown in Bessagnet *et al.* (2014) the large underestimation in 2009 are related to the underestimation of organics species. The observed diurnal cycles of $PM_{10}$ are very flat whatever the campaign with a small peak in the evening (Fig. 14). The systematic underestimation of $PM_{10}$ can be clearly observed but the shape of cycle is not very well captured, the evening peak is not reproduced. The models simulate low concentrations in the afternoon mainly driven by the elevation of the PBL. For the 2009 campaign, MINNI reproduces very well the diurnal cycle until 16:00. As shown in Fig. 15, dust concentrations

are higher for MINNI in the center of the domain. MINNI uses a parameterisation for wind blown dust very productive over any land cover types (Vautard *et al.*, 2005). In comparison EMEP mainly produces dust by traffic resuspension and a few over arable lands. This higher production of dust by MINNI in Europe certainly improve the PM negative bias usually observed in chemistry transport models and particularly in the afternoon when the wind are higher and the soil moisture lower.

Most of the underestimations of models is driven by too low day time $PM_{10}$ concentrations. It is noteworthy that MINNI calculate the lowest PBL that could explain this specific behaviour. For the summer campaign 2006, the $PM_{10}$ observations show an increase of concentrations in the afternoon while all models predict a decrease, indicating that all models are too sensitive to dynamical process (meteorology) and not sufficiently to the chemical formation.

**6.5    $PM_{2.5}$**

Performances on $PM_{2.5}$ are rather different compared to $PM_{10}$ (Fig. 5). MINNI generally shows a slight positive bias while all models underestimate the averaged concentrations, CMAQ having the highest negative bias. The performances of CHIMERE on the correlation are very good for all campaigns, its RMSE being the lowest for three campaigns. As for $PM_{10}$, the "ensemble" captures rather well the spatial patterns of $PM_{2.5}$. The concentrations in the south of Europe (Fig. 16 and

supplementary material S6) are not specifically underestimated except in Cyprus where dust events also contribute to increase the $PM_{2.5}$ concentrations. Whatever the campaign the coefficient of variation for $PM_{2.5}$ is the lowest in Spain but the RMSE of the "ensemble" is not particularly low in this region. The coefficient of variation is generally high over the north east part of the domain. For all campaigns the models simulate a hot spot over the north of Italy. As shown in the



supplementary material S6, CMAQ better than the other models captures the $PM_{2.5}$ concentrations in Ispra (IT04) for 2007 and 2008 campaigns, this station located at the border of the Po valley hot spot is usually underestimated by the models due to the very stable meteorology in this region. The spatial correlations are usually better for $PM_{2.5}$ for all models except for the summer campaign (Table 5).

As for the $PM_{10}$ concentrations, the diurnal cycle of $PM_{2.5}$ is rather flat with very small morning and evening peaks (Fig. 17). The models have a different behaviour; they simulate a sharp decrease of concentrations in the afternoon consistent with $PM_{10}$ diurnal cycles. This confirms the lack of secondary production during daytime. The chemical schemes for the production of organic matter are still incomplete for one main reason. As suggested by Jathar *et al.* (2014) a large part of the "unspeciated" fraction of organic species react and produces secondary organic matter and gasoline vehicles could be an

important contributor as well as wood burning emissions according Denier van der Gon *et al.* (2015). This unspeciated fraction is not included in our emission inventory explaining a part of the negative bias of models observed either in winter and summer campaigns particularly during the afternoon. This suggests that models with negative biases on $PM_{2.5}$ concentrations are coherent with the completeness of our inventory and the state-of-the-art of knowledge on SOA modelling.

## 7   Impact of meteorology on pollutant concentrations

### 7.1   Impact of the PBL parameterization with MINNI results for the 2009 campaign

As shown in the previous section, MINNI underestimates the PBL heights calculated at 12:00 from measurements but it is in a better agreement with hourly data available at SIRTA (Fig. 3). In order to test the effect of PBL heights on air quality predictions, the MINNI model has been run using the PBL from IFS instead of its own parameterization for PBL heights. As

Shown by Curci *et al.* (2015), processes in the PBL can greatly affect the $PM_{2.5}$ concentrations at the ground, for instance temperature and relative humidity can favour the production of ammonium nitrate in the upper PBL.

Fig. 18 shows the average PBL heights and the average concentrations of $O_3$, $NO_2$ and $PM_{10}$ using MINNI's parameterizations (left graphs) and the percentage difference between the average concentrations calculated with PBL heights given by IFS ($PBL_{IFS}$) and by MINNI's parameterizations ($PBL_{MINNI}$) (right graphs) using the following formula:

$(PBL_{IFS}-PBL_{MINNI})/PBL_{MINNI}$.

It can be seen that over the sea, on average, PBL heights calculated with MINNI's parameterizations ($PBL_{MINNI}$) are lower than PBL heights given by IFS ($PBL_{IFS}$) but over the land $PBL_{MINNI}$ is higher than $PBL_{IFS}$ in coastal areas, North Africa, Scandinavian mountains and middle of Russian plains, and lower over the rest. Over the sea, $PBL_{IFS}$ are higher than $PBL_{MINNI}$ more than 50% while over the land the differences are between -30 and +30%.

Fig. 18 also shows that the $O_3$ concentrations increase in correspondence of the increase of PBL heights up to 10% and more, and decrease where the PBL heights decrease. This behaviour is explained by the fact that with a higher PBL more $O_3$ is entrained from high altitudes where $O_3$ concentrations are higher than at surface. Since the $NO_2$ sources are mainly at





surface, the $NO_2$ concentrations generally decrease with the increase of PBL heights and increase with the decrease of PBL heights as a consequence of more or, respectively, less effective dilution. Over most of Europe, the $NO_2$ concentrations decrease up to 8% when $PBL_{IFS}$ heights are used. The $PM_{10}$ concentrations respond to PBL heights variation in the same way as $NO_2$. The use of $PBL_{IFS}$ heights produces a 4 % decrease of $PM_{10}$ concentrations in most parts of Europe but an increase

of 6-8% in coastal areas and Russian plains.

In terms of statistics, the use of the PBL from IFS in MINNI slightly improves the correlations mainly driven by an improvement of time correlations. $PM_{10}$, $PM_{2.5}$ and $NO_2$ concentrations are decreased by less than 0.5 µg m$^{-3}$, improving all error statistics reported in Fig. 5 for MINNI. An increase of 2.75 µg m$^{-3}$ is observed for $O_3$ concentrations. It is also worth to mention that the variations in pollutant concentrations are small (over the land below 10% generally) in comparison to the

variations of PBL height, therefore other factors such as emissions spatial distribution, meteorology (e.g. advection and vertical dispersion, especially in low-wind areas), gas phase chemistry, aerosol physics and chemistry have to be investigated for improving model performances.

These results clearly show the importance of having good estimates of PBL heights but they also demonstrates that more investigations are necessary in order to identify the best parameterization of PBL heights but also vertical diffusivities and

vertical advection schemes which improves the simulated concentrations over the whole Europe.

## 7.2    Influence of meteorology on $NO_2$ concentrations with CAMx results

Pollutant concentrations are strongly influenced by the reconstruction of meteorological fields. In this section a comparison of modelling performances in reproducing wind speed and $NO_2$ concentrations is presented and discussed. Furthermore,

Planetary Boundary Layer (PBL) height data, collected at SIRTA site (Paris) have been used too. Being mainly related to emission processes, $NO_2$ has been selected as a good tracer of the influence of dispersion on pollutant concentrations. The analysis has been performed over the Paris area since the hourly variation of the PBL is available. Two other smaller areas, namely: the whole Germany (DE), the Po Valley (POV) has been selected to complement the analysis.

$NO_2$ observed data set has been set up from AirBase database (Mol and de Leeuw, 2005), selecting just background stations,

having more than 75% valid data over the whole 2009. Finally, as already mentioned, PBL heights derived at SIRTA site has been included too. Modelled concentrations have been derived from the CAMx simulation results, while modelled meteorological fields have been derived from IFS.

In the case of the Paris area, the meteorological model showed a very good performance in reproducing the observed wind speed, whose temporal evolution clearly influences the corresponding temporal variability of $NO_2$ concentrations (Fig. 19).

Also the PBL height is quite well reproduced by the model, though the model tends to underestimate the night-time minima and, conversely, to overestimate some diurnal peaks.

Within the Paris area $NO_2$ observations are quite well reproduced by CAMx, showing a low bias of the median value lower than 2 ppb, corresponding to less than 20% of the observed median concentration (Fig. 19). The availability of both wind



speed and PBL height observations, allow the influence of both processes to be clearly detected. For example 3-4, 10 and 25 of March, the underestimation showed by CAMx seems well related to a corresponding overestimation of the PBL rather than the wind speed (Fig. 20). Conversely during night hours of March 5, CAMx results are more influenced by the wind speed.

The analysis has been completed comparing the diurnal cycle of both $NO_2$ and meteorological variables, reported in Fig. 21 and Fig. 22. At German sites $NO_2$ concentrations are slightly overestimated during night-time and underestimated during daytime. This behaviour does not seem strictly related to wind speed, particularly during night-time, thus being probably more related to vertical turbulence. At Po valley sites, $NO_2$ values are systematically underestimated, while wind speed is correctly reproduced, even partially underestimated during daytime hours. $NO_2$ modelled concentrations show a clear low

bias during night-time, probably related to an imprecise reconstruction of the strong stable conditions that characterize this area during the cold season. The difficulty of model is enhanced during the morning hours, when the model is not able to capture the strength of the observed peak. The discrepancy is probably caused by a too rapid growth of the PBL during the first daytime hours. Late in the afternoon the $NO_2$ bias tends to decrease, probably thanks to a very quick collapse of PBL height after sunset.

At Paris sites, $NO_2$ modelled concentrations show a behaviour similar to the Po valley area. The availability of both wind speed and PBL height observations, allows most of the previous comments to be confirmed. Particularly it is worth noting that at SIRTA site, PBL height shows a too rapid increase during morning hours followed by a too strong decrease just after sunset.

**8    Discussion and Conclusions**

One of the main outcomes of such a multi-seasonal intercomparison is that in most cases model performances are more influenced by the model setup than the season. For example, CMAQ shows the worst RMSE for $NO_2$ over all campaigns, LOTOS-EUROS shows the lowest RMSE for $SO_2$ over all campaigns, conversely CAMx always exhibits the highest RMSE for $SO_2$ over all periods. This means that in several case either the model formulation or the input setup influence the model

performance more than specific features of the meteorological season.

Whatever the pollutant and the campaign, there is not a strong correlation between the performances of the *ensemble* (through the RMSE) with the variability (coefficient of variation) of models. This means if models are close between each other, the mean of models can be far or close to the observed values. However, for $SO_2$ and $PM_{2.5}$ a correlation of -0.2 to -0.3 is observed for three campaigns meaning that a large variability tends to improve the performance of the ensemble for these

compounds. The coefficient of variation is the lowest for ozone (below 10%) particularly in the afternoon hours (see supplementary material S7) and for the summer period 2006, while for $SO_2$ this coefficient is the highest generally between 30 and 40%. For PM this coefficient is about 10 to 20%, over several countries, the coefficient of variation is higher in the



afternoon highlighting the difference between chemical schemes for the aerosol chemistry more active during day times, conversely, the low coefficient of variability for $O_3$ confirms a coherence of ozone chemistry scheme between models.

Another general outcome stemming from the whole exercise is that model performances are more different from a pollutant to another than for the same pollutant within the different season. This confirms once again that on average model

formulation and setup are more influencing than meteorological conditions on model performance. One of the few exceptions is shown by $O_3$ in 2009 where model results were characterized by RMSE values very similar to the other years, whereas bias was negative instead of positive as in the three previous years. But, as already pointed out, such a result was mainly driven by a relevant underestimation in the ozone boundary concentrations from MACC.

The intercomparison proved that CTMs are affordable in reproducing ozone concentrations, showing an average RMSE

value corresponding to 30% of the mean observed concentration for daily values. Modelled daily cycles are generally more spread during nigh-time than daytime hours. This means that, though most models shared the same meteorology, including PBL height, they proved to be very sensitive to vertical dispersion and deposition parameterization, the two key processes governing $O_3$ concentration during night-time. During daytime modelled concentrations are more overlapping and showing a different ranking with respect to night hours. This means, as expected, that during daytime vertical mixing reconstruction is

more similar among models and chemical schemes exhibit a different efficiency in ozone production. This behaviour is not detectable in 2007, that was a cold and windy period, hampering the development of photochemical processes.

$NO_2$ performances are less robust than $O_3$. The RMSE represents about 70% of the observed mean concentration, but the value is even higher in case of CMAQ. Bias is negative for most models, except CMAQ, adopting a different meteorology and MINNI, characterized by lower PBL heights. CHIMERE biases are closer to 0 than other models sharing the same

meteorology, such as CAMx.

The Normalized RMSE of the models "ensemble" is characterized by a relevant spatial variability, proving that local emission sources and meteorological conditions strongly influence $NO_2$ performance. Likewise ozone, most of the discrepancies among models and with respect to observations take place during night-time, when the atmosphere is more stable. As most models share the same wind fields, the modelled spread in night-time concentrations can be related to

vertical dispersion. Such spread can be considered as a measure of the uncertainty related do vertical mixing and qualitatively correspond to 80-100% of the observed mean concentration. Daytime modelled concentrations are more similar among models and generally underestimated, though the modelled PBL field at noon seemed lower than the observed one. As already mentioned such a systematic discrepancy could be related to a measurement artefact, but also to photochemistry that could give rise to an excess of nitric acid. More accurate observations of Nitric acid and Nitrate would be required.

$SO_2$ shows the worst performance, with RMSE values corresponding to 130-160% of the observed mean concentrations. Highest errors are shown by CAMx, MINNI and RCG, they were characterized by lower PBL heights (for MINNI) and wind speed (for RCG) than other models, for CAMx the high errors cannot be explained at this stage. It is worth noting that the modelled diurnal cycles show a weak morning peak, more typical of surface sources not observed in measured data. Conversely, measured data present a diurnal peak, usually related to enhance downward mixing of aloft sources, where most




of SO$_2$ is emitted. Discrepancies among models and with respect to observations can also rely on chemistry. For example in 2009, Bessagnet *et al.* (2014) reports for CHIMERE an underestimate of SO$_2$ concentrations on hourly basis, while sulphate is overestimated; conversely RCG, adopting a more simplified approach for sulphur chemistry than CHIMERE, overestimates SO$_2$, while underestimates sulfates.

PM$_{10}$ models performances are less homogenous within the four years than other pollutants. Years 2006 and 2007 that were characterized by a more dispersive atmosphere show a mean RMSE around 10 µg m$^{-3}$, representing 55-65% of the mean observed concentration. Differently, the RMSE rises up 15 µg m$^{-3}$, representing more than 80% of the observed mean. The bias is best reproduced by EMEP and MINNI, while CAMx and CMAQ show the strongest underestimation. The analysis of each PM compound for season 2009 (Bessagnet *et al.*, 2014) revealed that MINNI and EMEP were characterized by rather
different scores, suggesting that their overall performance is influenced in a different way by both chemistry and meteorology. Particularly MINNI performance seem more driven by a reduced dispersion often giving rise to higher concentrations than other models, while EMEP seems more able to capture the evolution of the single PM compound. CAMx and CMAQ often show the strongest negative bias. As for CAMx this result is probably driven by the combined effect of meteorology (also NO$_2$ is underestimated by CAMx) as well as the absence of some key processes such as sea salt
and dust resuspension and a PM coarse chemistry. Differently CMAQ model was characterized by very high NO$_2$ concentrations putting in evidence a less dispersive atmosphere than other models. As for CMAQ, the low PM$_{10}$ values can probably related to deposition processes. Indeed, for 2009 episode (Bessagnet *et al.*, 2014) CMAQ proved to be more efficient than the other models for dry deposition of both NO$_X$ and SO$_X$ compounds.

The observed diurnal cycles of PM$_{10}$ are very flat whatever the campaign with a small peak in the evening. The PM$_{10}$
observations show an increase of concentrations in the afternoon while all models predict a decrease, indicating that all models are too sensitive to dynamical process (meteorology) and not sufficiently to the chemical formation. The analysis of individual compounds of PM will bring more detailed, it will be investigated in a companion paper.

Models performance for PM$_{2.5}$ is on average slightly better than PM$_{10}$, both in terms of bias and correlation. PM$_{2.5}$ concentration is less affected by natural processes, which are more relevant for coarse PM, therefore the obtained results
suggest that modelling natural processes still present some relevant weaknesses (Bessagnet *et al.*, 2014). Modelled diurnal cycles show improved performance in terms of bias, but not with respect to the daily evolution. Firstly, this result confirms that there are processes mainly affecting the coarse fraction that are still missing in state of art CTM, highlighted by the different bias between PM$_{10}$ and PM$_{2.5}$. Secondly, the differences in the daily pattern, particularly evident in 2006 where photochemistry is at its maximum, confirm that dilution processes during daytime hours are too efficient with respect to
chemical processes, thus preventing the increase of modelled concentrations during afternoon hours.

Even if the meteorology was prescribed in the exercise, some variables related to dispersion modelling such as the vertical diffusion and the PBL height are often diagnosed in the model pre-processing. This step involves important differences in the dispersion as was shown for a tracer species like CO. Although most models used the same PBL from IFS (CHIMERE, CAMx, LOTOS-EUROS, RCG), the variability of models PBL (including other PBL parameterisation as used in EMEP,



CMAQ and MINNI) shows important differences of PBL calculations over the ocean and the Mediterranean sea. IFS wind speeds are overestimated with a bias reaching 1 m s$^{-1}$, which can have a dramatic effect at low wind speed conditions.

The comparison of the meteorological fields pointed out that the reconstruction of the meteorological variables is still affected by relevant uncertainties. Wind speed simulated by IFS and COSMO showed a systematic difference along the whole day, with IFS providing an average wind speed that in 2007 and 2009 was 12% higher than COSMO. PBL reconstruction showed an even higher variability with a spread among the models corresponding to 27-29% of the mean midday PBL value of each campaign.

Some additional analyses with respect to meteorology have been carried out. As a first step, a sensitivity analysis with respect to PBL height was performed with MINNI model. Over the sea, $PBL_{IFS}$ are higher than $PBL_{MINNI}$ more than 50% while over the land the differences are between -30 and +30%. As a consequence, $O_3$ concentrations increase in correspondence of the increase of PBL heights up to 10% and more, due to enhanced entrainment and reduced $NO_X$ titration. Over most of Europe, the $NO_2$ concentrations decrease up to 8% when $PBL_{IFS}$ heights are used and the $PM_{10}$ concentration decreases by 4 % but also increases of 6-8% in coastal areas and Russian plains, where IFS PBL were lower than MINNI PBL. The PBL explain only a part of the overestimation of primary species but a complementary study has to be performed including the deposition processes.

A comparison of modelling performances in reproducing wind speed and $NO_2$ concentrations was performed too, also including some analysis of the influence of Planetary Boundary Layer (PBL) height estimation. The comparison of modelled concentrations against wind speed and PBL heights confirmed that meteorology strongly influences CTMs performance. Particularly the temporal evolution of wind speed is most responsible of model skilfulness in reproducing the daily variability of pollutant concentrations (*e.g.* the development of peak episodes), while the reconstruction of the PBL diurnal cycle seems more influencing in driving the corresponding pollutant diurnal cycle and hence the presence of systematic positive and negative bias detectable on daily basis.

To complement the analysis, companion papers will focus on depositions of sulphur/nitrogen compounds and on the behaviour of models for particulate matter species. This ensemble of analyses will help to prioritize the improvement of air quality models used in the frame of the CLRTAP.

**Acknowledgements**

The EBAS database has largely been funded by the CLRTAP-EMEP programme, AMAP and by NILU internal resources. Specific developments have been possible due to projects like EUSAAR (EBAS web interphase), EBAS-Online (upgrading of database platform) and HTAP (import and export routines to build a secondary repository for in support of www.htap.org. A large number of specific projects have supported development of data and metadata reporting schemes in dialog with data providers (CREATE, ACTRIS and others). For a complete list of programmes and projects for which EBAS serves as a database, please consult the information box in the Framework filter of the web interface. These are all highly acknowledged for their support particularly Anne-G. Hjellbrekke. INERIS was financed by the French Ministry in charge of Ecology. RSE




contribution to this work has been financed by the Research Fund for the Italian Electrical System under the Contract Agreement between RSE S.p.A. and the Ministry of Economic Development - General Directorate for Nuclear Energy, Renewable Energy and Energy Efficiency in compliance with the Decree of March 8, 2006. PSI contribution in this work was funded by the Swiss Federal Office for the Environment (FOEN).

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



List of figures

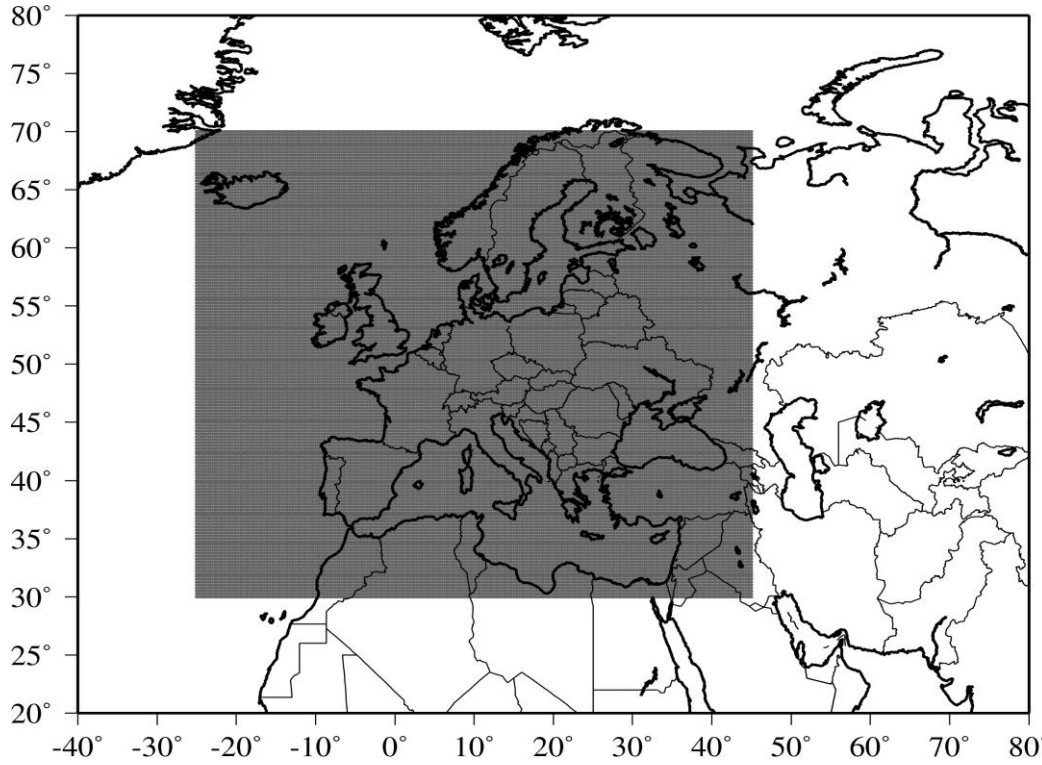

5    Fig. 1: The grey zone corresponds to the EURODELTA domain. All model simulations have been performed over this domain
except CMAQ.

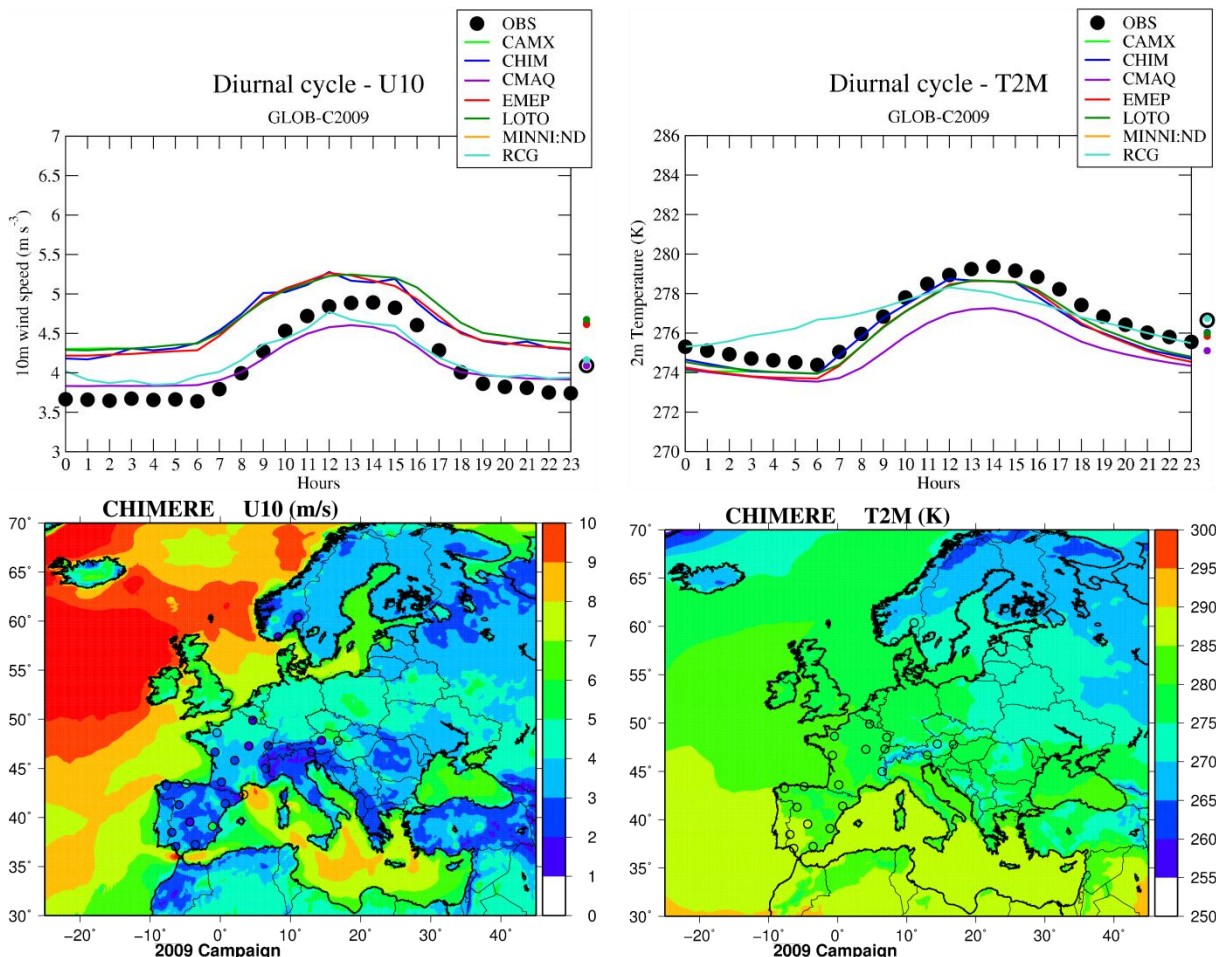

**Fig. 2: Comparisons of observed *versus* predicted meteorological variables (U10, T2M) for the 2009 campaign. *Top left panel*: mean diurnal cycle of the 10 m wind speed, *top right panel*: mean diurnal cycle of the 2 meter temperature, *bottom left panel*: mean 10 meters wind speed for CHIMERE, *bottom right panel*: mean 2 meters temperature for CHIMERE (Some observations at EMEP stations are provided with coloured circles over the maps). Red color is assigned for values exceeding the colour scale.**




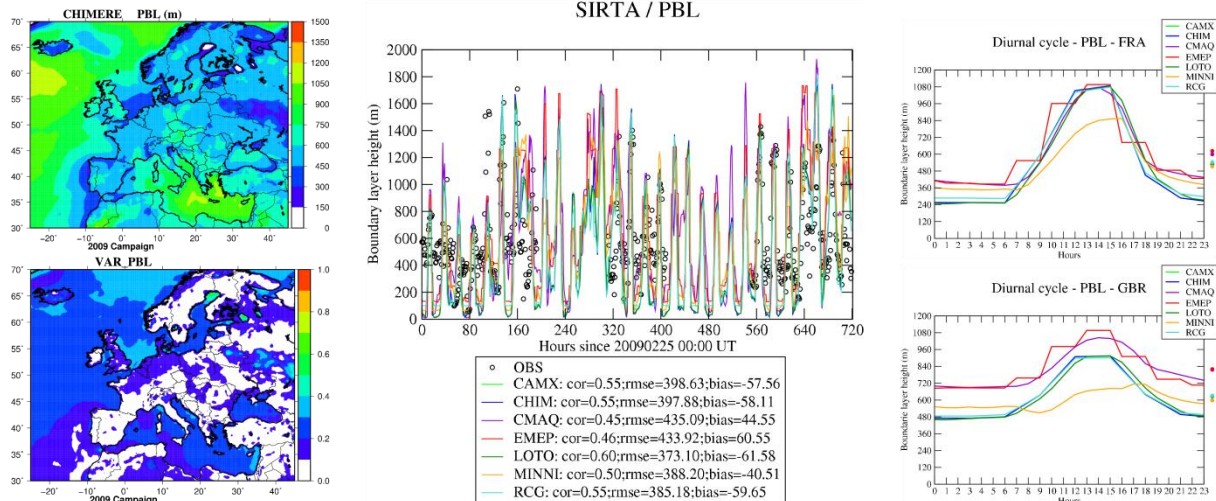

**Fig. 3:** Spatial representations and time variations of the PBL height for the 2009 campaign. *Top left panel:* Mean height of the CHIMERE PBL height issued from ECMWF data. *Bottom left panel:* Mean coefficient of variation for the PBL height. *Central panel:* hourly variation of the PBL height at the SIRTA station. *Top right panel*: Average diurnal cycle of the PBL height predicted by the models in France. *Bottom right panel*: Average diurnal cycle of the PBL height predicted by the models in Great Britain.





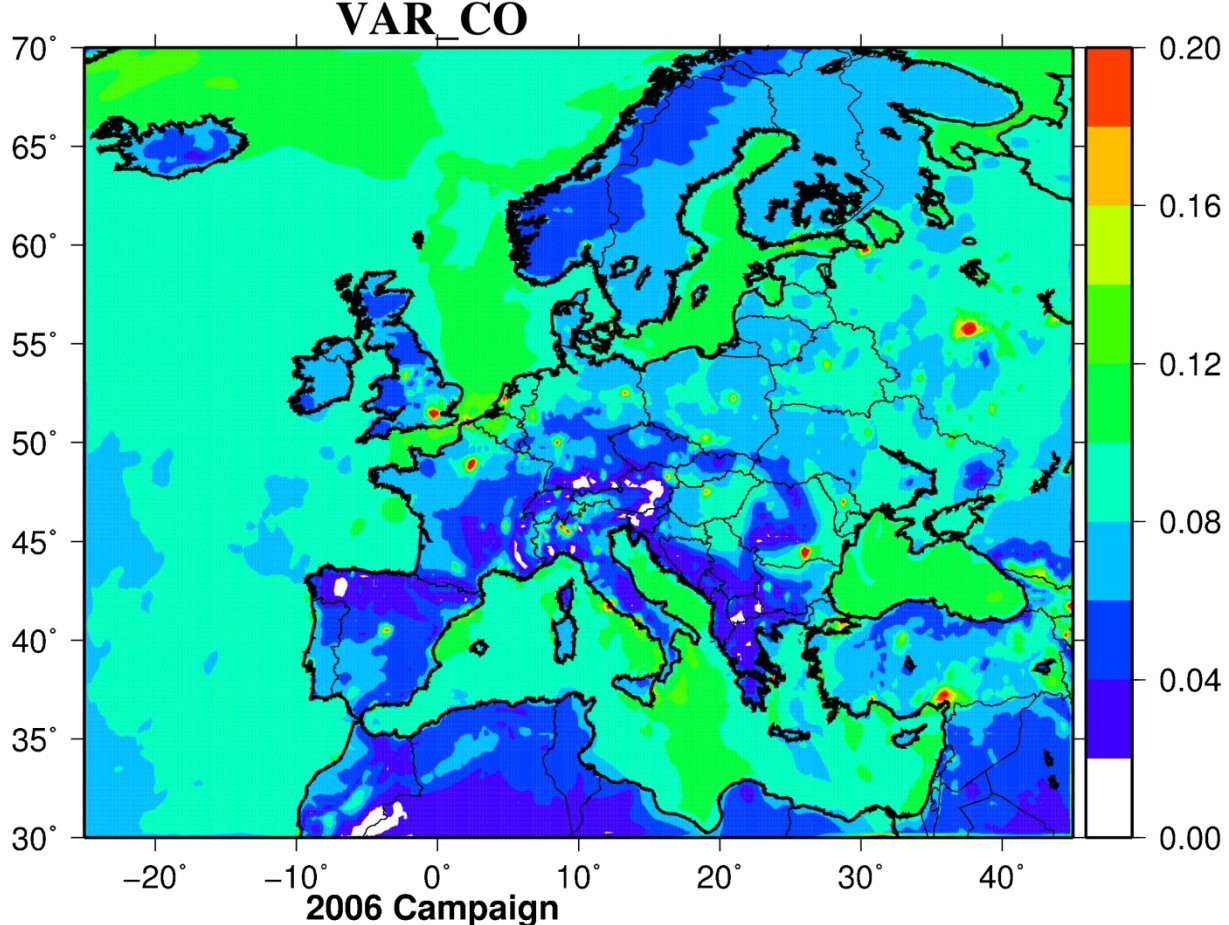

Fig. 4: Mean coefficient of variation of the CO concentrations predicted by the models for the 2006 campaign (no unit). Red color is assigned for values exceeding the color scale.





**Fig. 5: Overall performance of models for Ozone, Nitrogen dioxide, Sulphur dioxide, PM₁₀ and PM₂.₅ daily mean concentrations for all campaigns.**





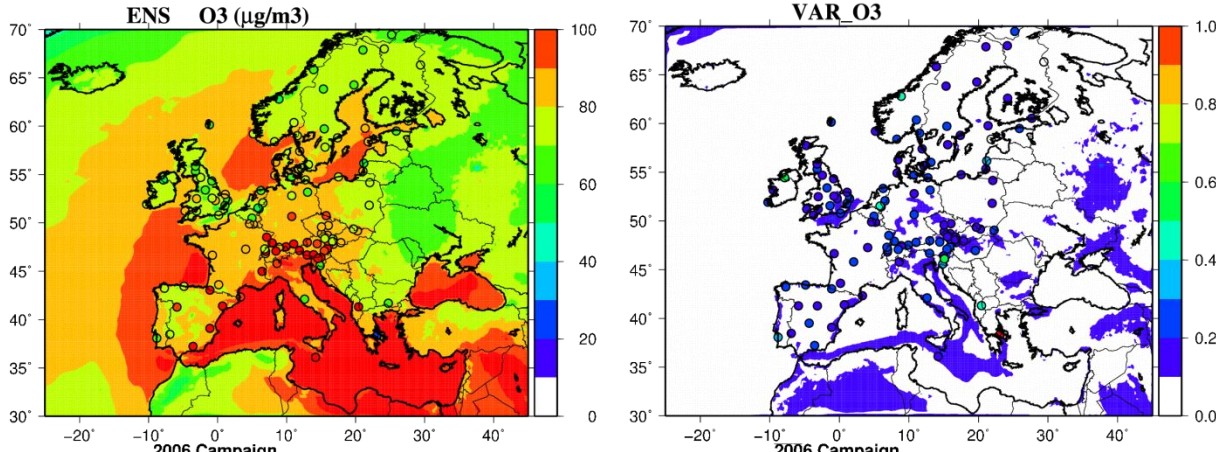

**Fig. 6:** *Left column*: **Mean ozone concentrations (μg m$^{-3}$) of the "ensemble" (ENS) for the 2006 campaign with corresponding observations (coloured dots).** *Right column*: **coefficient of variation of models (no unit) constituting the ensemble with corresponding normalized root mean square errors of the "ensemble" (coloured dots). Red color is assigned for values exceeding the color scale.**



**Fig. 7: Mean ozone diurnal cycles for all campaigns simulated by the models compared with observations. Averaged concentrations are provided on the right side of the charts.**





**Fig. 8: Timeseries of hourly concentrations at Mace Head for all models and campaigns**





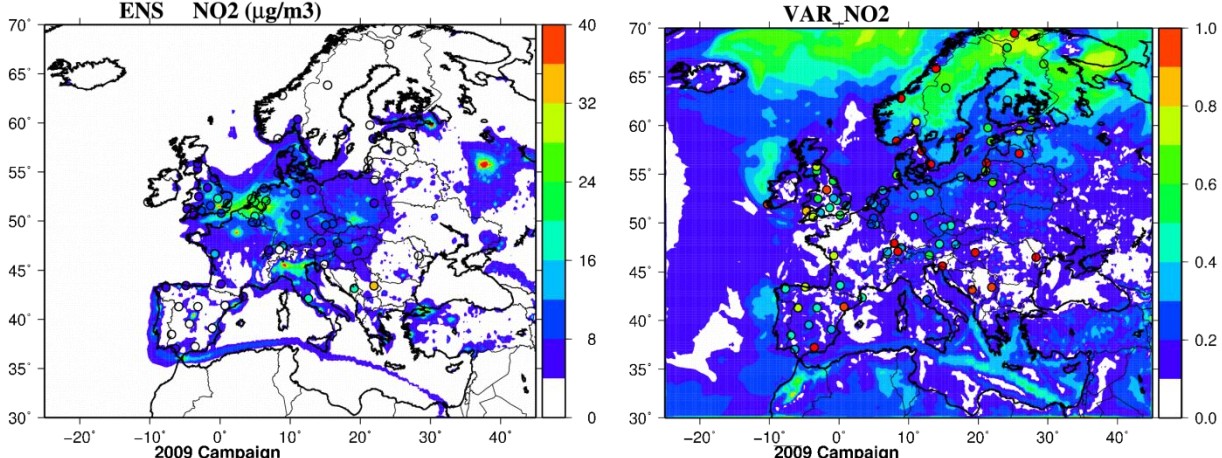

**Fig. 9:** *Left column*: **Mean nitrogen dioxide concentrations (µg m$^{-3}$) of the "ensemble" (ENS) for the 2009 campaign with corresponding observations (coloured dots).** *Right column*: **coefficient of variation of models (no unit) constituting the ensemble with corresponding normalized root mean square errors of the "ensemble" (coloured dots). Red color is assigned for values exceeding the color scale.**







**Fig. 10: Mean diurnal cycles of nitrogen dioxide for all campaigns simulated by the models compared with observations. Averaged concentrations are provided on the right side of the charts.**





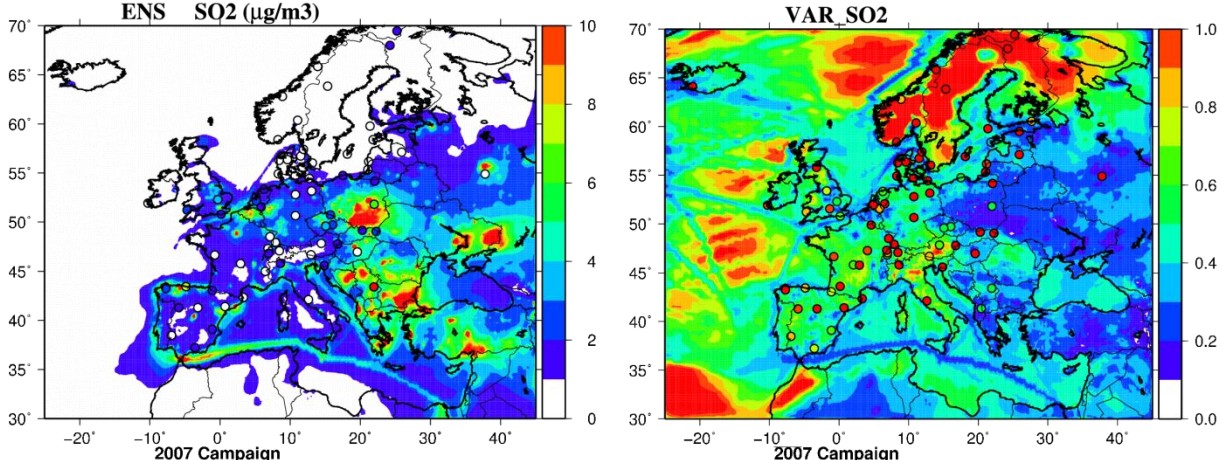

**Fig. 11:** *Left column*: **Mean SO$_2$ concentrations (µg m$^{-3}$) of the "ensemble" (ENS) for the 2007 campaign with corresponding observations (coloured dots).** *Right column*: **coefficient of variation of models (no unit) constituting the ensemble with corresponding normalized root mean square errors of the "ensemble" (coloured dots). Red color is assigned for values exceeding the color scale.**





**Fig. 12: Mean SO2 diurnal cycles for all campaigns simulated by the models compared with observations. Averaged concentrations are provided on the right side of the charts.**



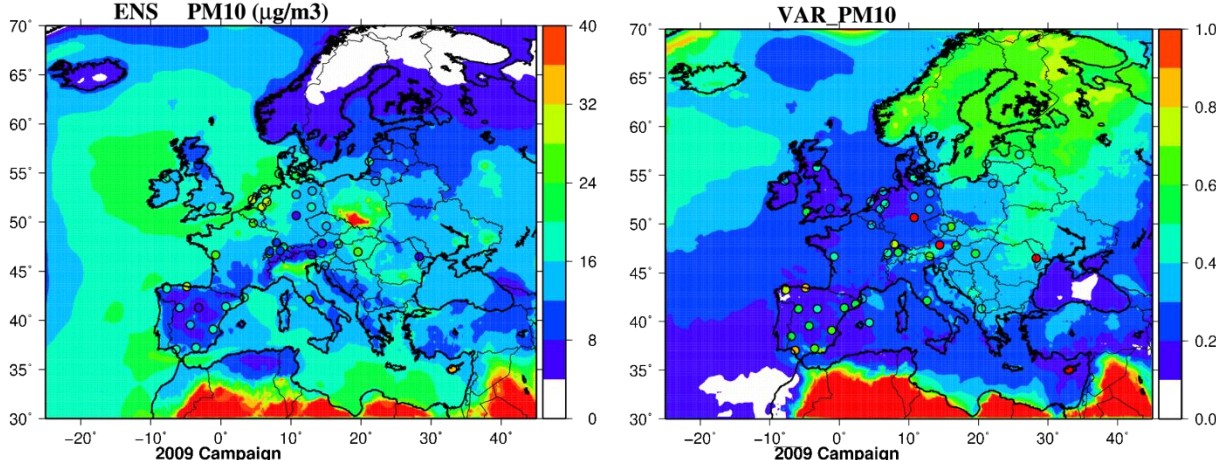

**Fig. 13:** *Left column*: Mean PM$_{10}$ concentrations (µg m$^{-3}$) of the "ensemble" (ENS) for the 2009 campaign with corresponding observations (coloured dots). *Right column*: coefficient of variation of models (no unit) constituting the ensemble with corresponding normalized root mean square errors of the "ensemble" (coloured dots). Red color is assigned for values exceeding the color scale.



**Fig. 14: Mean diurnal cycles of PM$_{10}$ for all campaigns simulated by the models compared with observations. Averaged concentrations are provided on the right side of the charts.**



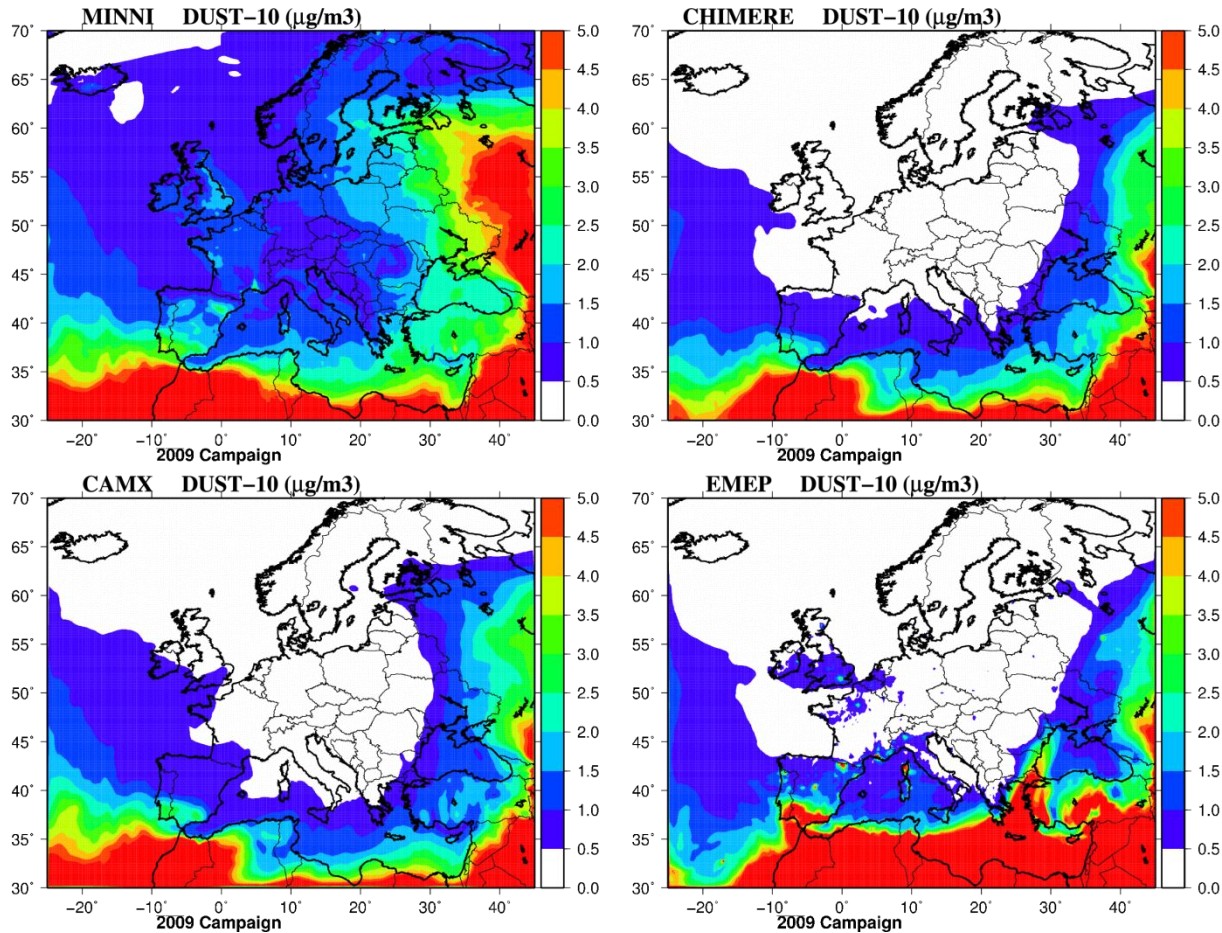

**Fig. 15: Mean dust concentrations (µg m⁻³) in the PM₁₀ fraction for the 2009 campaign computed by the MINNI, CHIMERE, CAMx and EMEP models.**





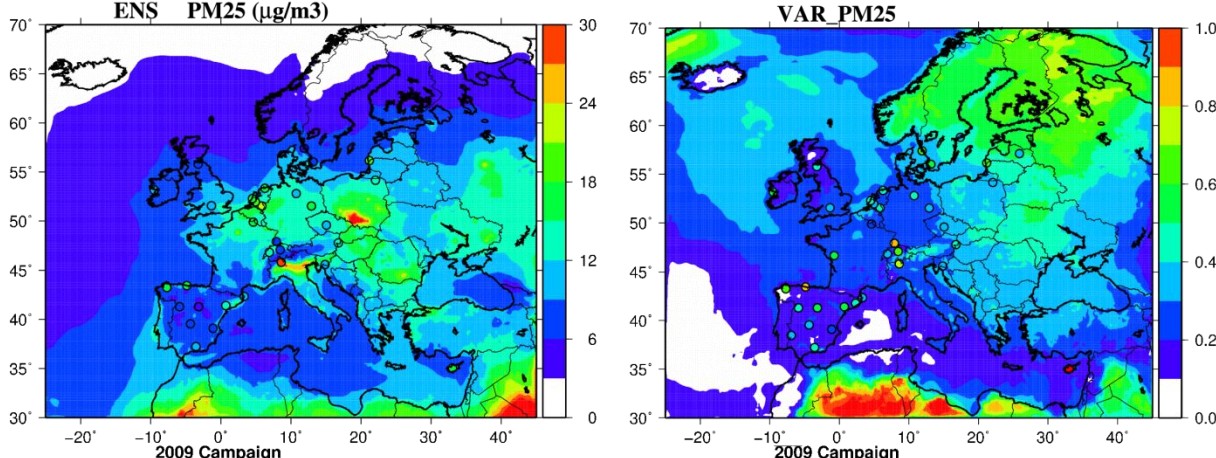

**Fig. 16:** *Left column*: **Mean PM$_{2.5}$ concentrations (µg m$^{-3}$) of the "ensemble" (ENS) for the 2009 campaign with corresponding observations (coloured dots).** *Right column*: **coefficient of variation of models (no unit) constituting the ensemble with corresponding normalized root mean square errors of the "ensemble" (coloured dots). Red color is assigned for values exceeding the color scale.**





**Fig. 17: Mean diurnal cycles of PM$_{2.5}$ for all campaigns simulated by the models compared with observations. Averaged concentrations are provided on the right side of the charts.**





**Fig. 18: Left graphs show the average PBL heights and the average concentrations for O₃, NO₂ and PM₁₀ using original MINNI's parameterizations. Right graphs show the percentage difference between the average concentrations calculated with PBL heights given by IFS (PBL$_{IFS}$) and by MINNI's parameterizations (PBL$_{MINNI}$). Red color is assigned for values exceeding the color scale.**



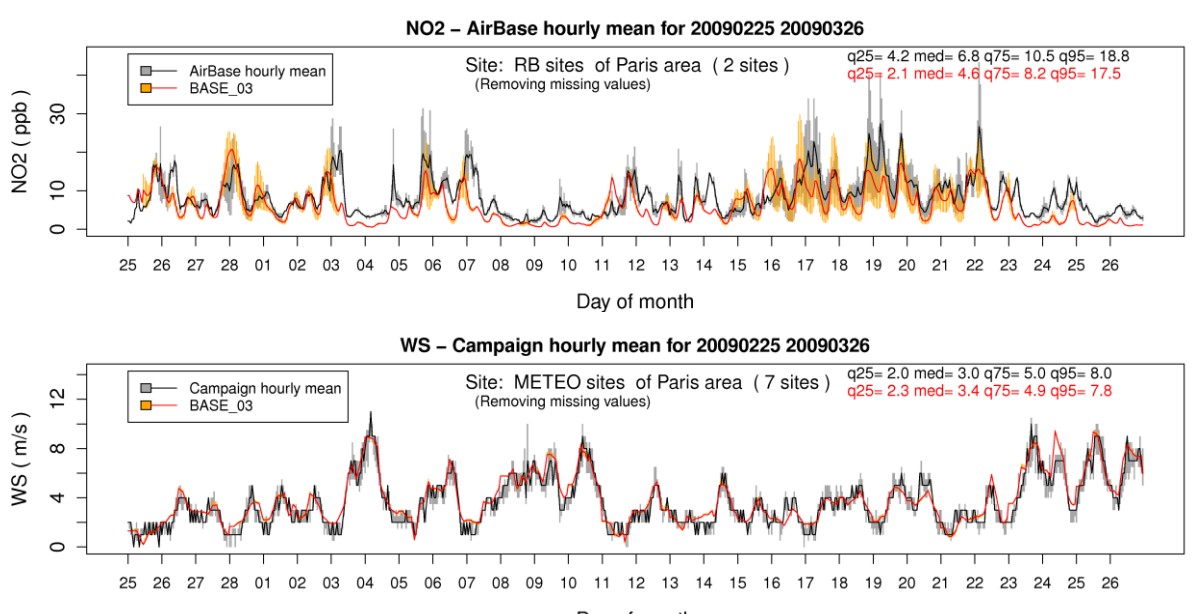

**Fig. 19:** Time series of hourly Box plots showing the distribution of the observed and computed NO₂ concentration (top) and wind speed (bottom) for CAMx (meteorology from IFS). Observations are in black/grey; modelled values in red/orange. Bars show the 25th -75th quantile interval, while the median is displayed by the continuous line. The 25th, 50th, 75th, and 95th quantile of the whole campaign are reported too. Comparison of computed and observed boxplot time series evaluated at Airbase and meteorological sites, available over the Paris area.

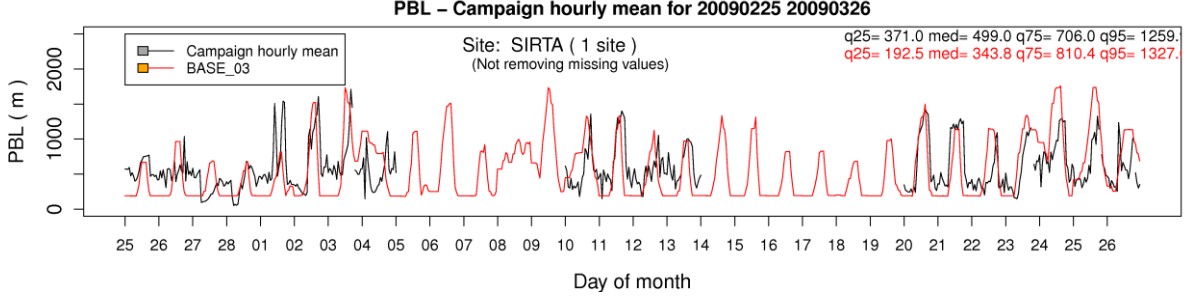

**Fig. 20:** Time series of hourly Box plots showing the distribution of the observed and computed PBL height. Observations are in black/grey; modelled values in red/orange. Bars show the 25th -75th quantile interval, while the median is displayed by the continuous line. The 25th, 50th, 75th, and 95th quantile of the whole campaign are reported too. Comparison of computed and observed boxplot time series evaluated at SIRTA site.





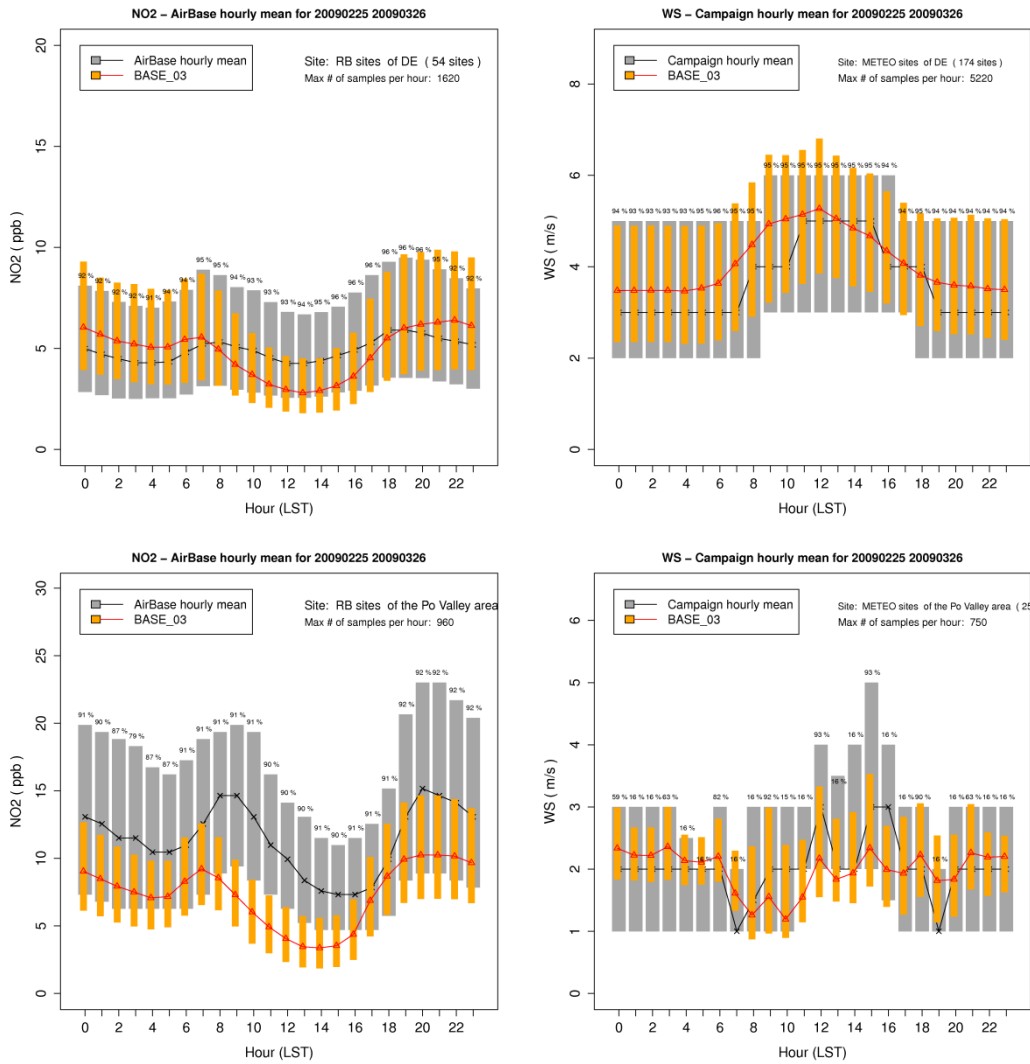

5   **Fig. 21: Time series of hourly Box plots showing the distribution of the diurnal cycle observed and computed NO$_2$ concentration (left) and wind speed (right) over Germany (top panels) and Po valley (bottom panels). Observations are in black/grey; modelled values in red/orange. Bars show the 25$^{th}$ -75$^{th}$ quantile interval, while the median is displayed by the continuous line. The 25$^{th}$, 50$^{th}$, 75$^{th}$, and 95$^{th}$ quantile of the whole campaign are reported too. Comparison of computed and observed boxplot time series evaluated at AirBase and meteorological sites, available over Germany and Po valley. Hour is in UTC time.**



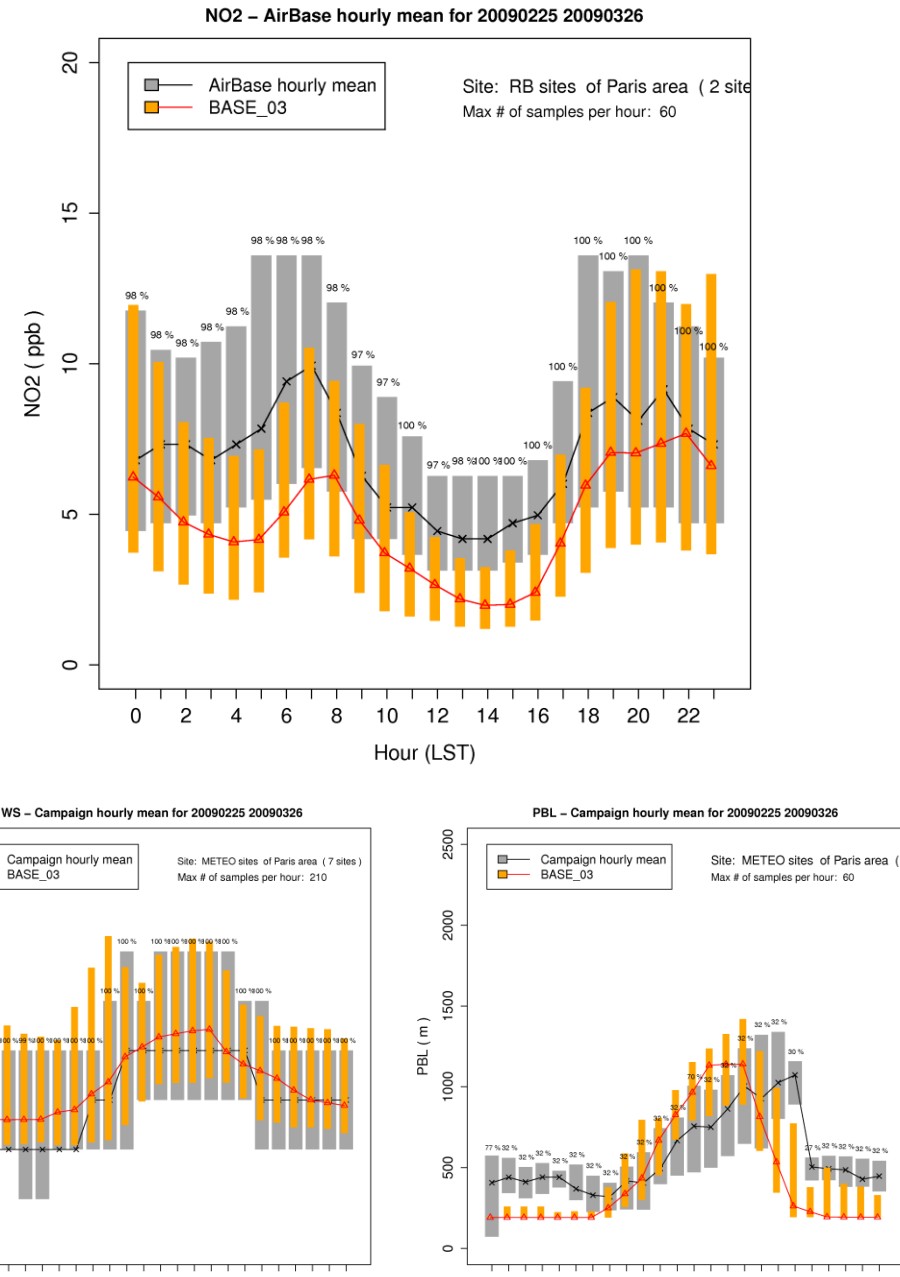

**Fig. 22: Time series of hourly Box plots showing the distribution of the diurnal cycle observed and computed NO$_2$ concentration (top), wind speed (bottom left) and PBL height (bottom right). Observations are in black/grey; modelled values in red/orange. Bars show the 25$^{th}$ -75$^{th}$ quantile interval, while the median is displayed by the continuous line. The 25$^{th}$, 50$^{th}$, 75$^{th}$, and 95$^{th}$ quantile of the whole campaign are reported too. Comparison of computed and observed boxplot time series evaluated at AirBase and meteorological sites, available over the Paris area. Hour is in UTC time.**





**List of tables**

**Table 1: Models involved in the study**

| Teams | Models with references | Model acronym in this study | Simulated periods |
|---|---|---|---|
| PSI/RSE | **CAMx** (ENVIRON, 2003) | CAMX | 2006, 2007, 2008, 2009 |
| INERIS | **CHIMERE** (Menut *et al.*, 2013) | CHIM | 2006, 2007, 2008, 2009 |
| HZG | **CMAQ** (Byun *et al.*, 2006; Matthias et al., 2008) | CMAQ | 2006, 2007, 2008, 2009 |
| MSC-W - Met.NO | **EMEP** (Simpson *et al.*, 2012) | EMEP | 2006, 2007, 2008, 2009 |
| TNO | **LOTOS-EUROS** (Sauter *et al.*, 2014) | LOTO | 2006, 2007, 2008, 2009 |
| ENEA/ARIANET | **MINNI** (ARIANET, 2004) | MINNI | 2006, 2007, 2008, 2009 |
| FUB | **RCG** (Stern *et al.*, 2006) | RCG | 2008, 2009 |



**Table 2: Synthetic description of models (part 1)**

| | EMEP | CHIMERE | LOTOS-EUROS | RCG | CMAQ | MINNI | CAMx |
|---|---|---|---|---|---|---|---|
| **version** | rv4.1.3 | Chimere2013 | v1.8 | v2.1 | V4.7.1 | FARM V3.1.12 | V5.40 |
| **VERTICAL MODEL STRUCTURE** | | | | | | | |
| **Vertical layers** | 20 sigma | 9 sigma | 4 (3 dynamic layers and a surface layer) | 6 fixed terrain following layers | 30 sigma | 16 fixed terrain-following layers | 33 sigma |
| **Vertical extent (hpa or m)** | 100 hPa | 500 hPa | 3500 m | 3000 m | 100 hPa | 10000 m | 8000 m |
| **First layer depth** | 90 m | 20 m | 25 m | 25 m | 42 m | 40 m | 20 m |
| **Correction of first level concentration** | Yes | No | Yes | No | No | No | No |
| **NATURAL EMISSIONS** | | | | | | | |
| **Biogenic VOC** | Based upon maps of 115 species from Koeble and Seufert (2001), and hourly temperature and light. See Simpson *et al.* (2012) | MEGAN model v2.04 | Based upon maps of 115 species from Koeble and Seufert (2001), and hourly temperature and light (Beltman *et al.*, 2013) | Based upon maps of 115 species from Koeble and Seufert (2001), and hourly temperature and light.using emissions factors of Simpson *et al.* (1999) | BEIS 3.14 emission inventory (Vukovich and Pierce, 2002) | MEGAN model v2.04 | MEGAN model v2.1 |
| **Soil NO** | After Simpson *et al.* (2012) | MEGAN model v2.04 | Not used here | From Simpson *et al.* (1999) | BEIS 3.1.4 | MEGAN v2.04 | MEGAN model v2.1 |
| **Lightning emissions** | Climatological fields, Köhler *et al.* (1995) | No | No | No | No | No | No |
| **Sea salt** | Monahan (1986) and Martensson (2003), see Tsyro *et al.* (2011). | Monahan *et al.* (1986) | Martensson *et al.* (2003) and Monahan *et al.* (1986) | Gong *et al.* (1997) and Monahan *et al.* (1986) | Zhang *et al.* (2005) and Clarke *et al.* (2006) | Zhang *et al.* (2005) | Not used |
| **Windblown Dust** | After Simpson *et al.* (2012) | No | Denier van der Gon *et al.* (2009). | Loosemore and Hunt (2000), Claiborn *et al.* (1998) | No | Vautard *et al.* (2005) | No |
| **Road traffic suspension** | Denier van der Gon *et al.* (2009). | No | No | No | No | No | No |
| **LANDUSE** | | | | | | | |
| **Landuse database** | CCE/SEI for Europe, elsewhere GLC2000 | GLOBCOVER (24 classes) | Corine Land Cover 2000 (13 classes) | Corine Land Cover 2000 (13 classes) | Corine Land Cover 2006 (44 classes) | Corine Land Cover 2006 (22 classes) | USGS data |
| **Resolution** | Flexible, CCE/SEI ~ 5 km | About 300 m | 1/60 x 1/60 degrees | 1/60 x 1/60 degrees | About 250 m | About 250 m | 10 minutes |



**Table 3: Synthetic description of models (part 2)**

| | EMEP | CHIM | LOTO | RCG | CMAQ | MINNI | CAMX |
|---|---|---|---|---|---|---|---|
| **METEOROLOGY** | | | | | | | |
| Driver | ECMWF IFS | ECMWF IFS + urban mixing | ECMWF IFS | ECMWF IFS + Observations | COSMO CLM | ECMWF IFS | ECMWF IFS |
| Resolution | 0.25°x0.25° | 0.25°x0.25° | 0.25°x0.25° | 0.25°x0.25° | 24 km x 24 km (Lambert Conformal Conic Projection) | 0.25°x0.25° | 0.25°x0.25° |
| **PROCESSES** | | | | | | | |
| Advection scheme | Bott (1989a,b) | Van Leer (1984) | Walcek (2000) | Walcek (2000) modified by Yamartino (2003). | Blackman cubic polynomials (Yamartino, 1993) | Blackman cubic polynomials (Yamartino, 1993) | Bott (1989a,b) |
| Vertical diffusion | Kz approach following O'Brien (1970) and on Jeričevič et al. (2010) for stable and neutral conditions | Kz approach following (Troen and Mart, 1986) IFS PBL | Kz approach IFS PBL | Kz-approach and IFS PBL | ACM2 PBL scheme (Pleim, 2007a) | Kz following Lange (1989). PBL from Maul (1980) version of Carson (1973) algorithm for day times. | Kz approach following O'Brien (1970) IFS PBL |
| Dry deposition scheme | resistance approach for gases, Venkatram and Pleim (1999) for aerosols, Simpson et al. (2012) | resistance approach Emberson (2000a,b) | Resistance approach,DEPAC3.1 1for gases, Van Zanten et al. (2010) and Zhang et al (2001) for aerosols | resistance approach, DEPAC-module | Resistance approach, Venkatram and Pleim (1999) | Resistance model (Walcek and Taylor, 1986; Wesely, 1989) | Resistance model for gases (Zhang et al.,2003) and aerosols (Zhang et al., 2001) |
| Compensation points | No, but zero $NH_3$ deposition over growing crops | No | Only for $NH_3$ (for stomatal, external leaf surface and soil = 0) | No | No | No | No |
| Stomatal resistance | DO3SE-EMEP: Emberson et al. (2000a,b), Tuovinen et al. (2004), Simpson et al. (2012) | Emberson (2000a,b) | Emberson (2000a,b) | Wesely (1989) | Wesely (1989) | Wesely (1989) | Wesely (1989) |
| Wet deposition of gases | In-cloud and sub-cloud scavenging coefficients | In-cloud and sub-cloud scavenging coefficients | sub-cloud scavenging coefficient | pH dependent scavenging coefficients | In-cloud and sub-cloud scavenging which depends on Henry's law constants, dissociation constants and cloud water pH. Chang et al. (1987) | In-cloud and sub-cloud scavenging coefficients (EMEP, 2003) | In-cloud and sub-cloud scavenging model for gases and aerosols (Seinfeld and Pandis, 1998) |
| Wet deposition of particles | In-cloud and sub-cloud scavenging | In-cloud and sub-cloud scavenging | Sub-cloud scavenging coefficient | Sub-cloud scavenging coefficients | In-cloud and sub-cloud scavenging | In-cloud and sub-cloud scavenging coefficients | In-cloud and sub-cloud scavenging model for gases and aerosols (Seinfeld and Pandis, 1998) |
| Gas phase chemistry | EmChem09 (Simpson et al. | MELCHIOR | TNO CBM-IV | CBM-IV | CB-05 with chlorine chemistry | SAPRC99 (Carter, 2000a,b) | CB-05 (Yarwood et al., 2005) |





| | | | | | | | extensions (Yarwood et al., 2005) |
|---|---|---|---|---|---|---|---|
| | 2012) | | | | | | |
| **Cloud chemistry** | Aqueous SO₂ chemistry | Aqueous SO₂ chemistry and ph computation | No | Simplified aqueous SO₂ chemistry | Aqueous SO₂ chemistry (Walcek and Taylor, 1986) | Aqueous SO₂ chemistry (Seinfeld and Pandis, 1998) | Aqueous SO₂ chemistry RADM-AQ (Chang et al., 1987) |
| **Coarse nitrate** | Yes | No reactions with Ca or Na but coarse might exist with transfer from smaller particles | Yes | Yes | No | No | No |
| **Secondary Inorganic equilibrium** | MARS (Binkowski and Shankar,1995) | ISORROPIA (Nenes et al., 1999) | ISORROPIA v.2 | ISORROPIA | ISORROPIAv1.7 | ISORROPIA v1.7 (Nenes et al., 1998) | ISORROPIA (Nenes et al., 1998) |
| **SOA formation** | VBS-NPAS – Simpson et al. (2012) | After Bessagnet et al. (2009) | Based on Bergström et al (2012) | SORGAM module (Schell et al., 2001) | SORGAM module (Schell et al., 2001) | SORGAM module (Schell et al., 2001) | CAMx-VBS (beta version) (Koo et al., 2014) |
| **VBS** | Yes, Bergström et al (2012), Simpson et al. (2012) | No | Yes, based on Bergström et al (2012) | No | No | No | Yes based on Koo et al. (2014) |
| **Aerosol model** | Bulk- approach (2 modes) | 8 bins (40 nm to 10 µm) | Bulk- approach (2 modes) | Bulk approach (2 modes) | AERO5 (Carlton et al., 2010), Log-normal approach (3 modes) | AERO3 (Binkowski, 1999); 3 modes: Aitken, accumulation, coarse | Bulk- approach (2 modes) |
| **Aerosol physics** | No dynamics | Coagulation/condensation/nucleation | No dynamics | No dynamics | Coagulation/condensation/nucleation | Coagulation/condensation/nucleation | No dynamics |



**Table 4: Error statistics used to evaluate model performance (M and O refer respectively with Model and Observations data, and N is the number of observations)**

| | |
|---|---|
| **Mean Bias** | $(\bar{M} - \bar{O})$ with $\bar{M} = \frac{1}{N}\sum_{i=1}^{N} M_i$ and $\bar{O} = \frac{1}{N}\sum_{i=1}^{N} O_i$ |
| **Normalised Mean Bias** | $NMB = (\bar{M} - \bar{O})/\bar{O}$ |
| **Mean Bias** | $MB = (\bar{M} - \bar{O})$ |
| **Mean Gross Error** | $MGE = \frac{1}{N}\sum_{i=1}^{N} |M_i - O_i|$ |
| **Standard Deviation** | $SD_X = \sqrt{\frac{1}{N}\sum_{i=1}^{N}(X_i - \bar{X})^2}$ with X=O or M |
| **Root Mean Square Error** | $RMSE = \sqrt{\frac{1}{N}\sum_{i=1}^{N}(M_i - O_i)^2}$ |
| **Normalized Root Mean Square Error** | $NMSE = \frac{1}{\bar{M}}\sqrt{\frac{1}{N}\sum_{i=1}^{N}(M_i - O_i)^2}$ |
| **Correlation Coefficient** | $R = \left(\sum_{i=1}^{N}(M_i - \bar{M})(O_i - \bar{O})\right)\bigg/\left(\sqrt{\sum_{i=1}^{N}(M_i - \bar{M})^2 \times \sum_{i=1}^{N}(O_i - \bar{O})^2}\right)$ |

5  **Table 5: PM$_{10}$ and PM$_{2.5}$ spatial correlations for all campaigns**

| | 2006 | | 2007 | | 2008 | | 2009 | |
|---|---|---|---|---|---|---|---|---|
| | **PM$_{10}$** | *PM$_{2.5}$* | **PM$_{10}$** | *PM$_{2.5}$* | **PM$_{10}$** | *PM$_{2.5}$* | **PM$_{10}$** | *PM$_{2.5}$* |
| **CAMx** | 0.58 | *0.32* | 0.24 | *0.60* | 0.32 | *0.47* | 0.07 | *0.46* |
| **CHIMERE** | 0.65 | *0.32* | 0.58 | *0.78* | 0.39 | *0.42* | 0.55 | *0.66* |
| **CMAQ** | 0.50 | *0.19* | 0.50 | *0.80* | 0.11 | *0.42* | 0.11 | *0.37* |
| **EMEP** | 0.75 | *0.24* | 0.56 | *0.62* | 0.34 | *0.48* | 0.68 | *0.61* |
| **LOTOS-EUROS** | 0.34 | *0.05* | 0.50 | *0.61* | 0.27 | *0.37* | 0.50 | *0.37* |
| **MINNI** | 0.61 | *0.43* | 0.55 | *0.58* | 0.20 | *0.45* | 0.32 | *0.51* |
| **RCG** | ND | *ND* | ND | *ND* | 0.62 | *0.32* | 0.44 | *0.36* |