# Peer review of "Presentation of the EURODELTA III inter-comparison exercise -Evaluation of the chemistry transport models performance on criteria pollutants and joint analysis with meteorology"

_Atmospheric Chemistry and Physics, 2015_

## Referee Comment (RC1) · Anonymous Referee #2 · 22 Apr 2016

Referee report regarding the manuscript:

**Presentation of the EURODELTA III inter-comparison exercise - Evaluation of the chemistry transport models performance on criteria pollutants and joint analysis with meteorology**

Authors: B. Bessagnet et al.

**General comments**

In my opinion this paper is not suitable for ACP — it is essentially a technical report from the EURODELTA project, and I think it should be published as such; a (rather confusingly presented) model intercomparison, for a limited set of "standard" atmospheric components, is not interesting enough for publication in ACP. Many similar model intercomparisons have been published and, as written, this one does not contribute anything new.

The paper only documents model results for seven different chemical transport models without enough detail to be able to draw any useful conclusions for the general scientific community. I can not see what this paper contributes to the understanding about the atmospheric chemistry or physics, or any new information that aids in improving modelling of the atmospheric composition.

I also think that the authors are fragmenting their research (over several papers). This should be avoided (see the ACP Obligations for authors). In order to better understand model performance you need to take into account all important processes — including deposition and chemistry. If the authors want to publish this material in a scientific journal I think it has to be combined with the information about deposition and chemical composition of particulate matter. Splitting the model evaluation into three different papers is not appropriate.

The paper could have been acceptable for Geoscientific Model Development (GMD, which accepts model evaluation papers) — if the presentation had been better — but I think that a much more scientific approach is needed to make the material presented in this manuscript interesting enough for publication in a peer-reviewed journal.

I am sure that there are a number of interesting scientific questions that the EURODELTA project can answer, and I suggest that the authors focus mainly on that, and keep this kind of model intercomparison/evaluation documentation to technical reports. This paper in itself has little significance for the ACP community.

The paper has a very long author list with 36 authors! However, the brief statement on page 3, about what seven of the participating institutes (and I think NILU is missing in this list) have contributed to the project, is not motivation enough for the inclusion of so many authors. Considering the very long author list, please give a brief explanation of what each individual has contributed to this paper in the reply to this referee comment (the statement of contributions can be added to the supplement of the paper). Please note Point 9 under the General Obligations for Authors for ACP (my highlighting):

"To protect the integrity of authorship, **only persons who have significantly contributed to the research and paper preparation should be listed as authors**. The corresponding author attests to the fact that any others named as authors have seen the final version of the paper and have agreed to its submission for publication." … "The author who submits a manuscript for publication accepts the responsibility of having included as co-authors all persons that are appropriate and none that are inappropriate."

**Specific comments**

Page 3, lines 34–35: "As a consequence, there were very limited differences in the models set up, representing a sort of sensitivity analysis to several aspects of the modelling chains."
- I do not understand what you mean by this sentence! What do you mean by "a sort of sensitivity analysis" and in what sense is this study a sensitivity analysis? I suggest that this sentence is removed.

Page 4, lines 3–4: "Complementary analyses of depositions fluxes and PM composition data at high temporal resolution will be discussed in companion papers in order to better understand the behaviour of models."
- In my opinion this is fragmentation of research papers, and the consequence is that the present paper becomes uninteresting. As mentioned in the General Comments, I do not think that splitting the information this way in three different papers is useful. If knowledge about the deposition fluxes and PM composition data are important for understanding the behaviour of the models (which I certainly expect them to be) this information is needed in the present paper!

Page 4, lines 25–26: "In CMAQ additional anthropogenic dust is calculated as 90% of unspecified PM coarse emissions and attributed to fugitive dust"
- What is the motivation for adding extra anthropogenic dust? Was this just a modelling mistake?
Or do you have good reasons to believe that the emission inventory used in the present study lacks a substantial amount of fugitive dust? And if this is the case, why did you not increase the emissions in all models?

Page 4, line 27: "CAMx did not activate the sea salts parameterisation in this exercise."

- Why not? Was this a modelling mistake? Or are there problems with the sea salt emissions in CAMx?

Page 4, lines 30–31: Why was CAMx not included in the ENSEMBLE for $O_3$, $NO_2$, and $SO_2$? I would guess that the lack of sea salt would hardly have any impact on these three gaseous species.

Page 8 — Emissions: Are the emissions used in the EURODELTA project available for use by scientists outside the project? If they are, please specify this and where they can be found. If they are not available, more details are needed regarding the emissions in order for others to be able to evaluate/compare this work to other studies using other emission data. Without more detailed information the work presented in this manuscript can not be considered reproducible.

Page 8, lines 16–22: "EMEP national emissions were kept except for..." 14 countries, for which GAINS emissions were used.
- This seems a bit strange - why did you change emissions for these 14 countries and not for the other countries? Please give a motivation.

"Additional factors were applied on two Polish regions (x4 or x8) for $PM_{2.5}$ and $PM_{10}$ emissions"
- For which Polish regions? They need to be specified in detail to make this work reproducible. It is also unclear if the x4 factor applies to $PM_{2.5}$ and x8 to $PM_{10}$ or if the same factors were used for $PM_{2.5}$ and $PM_{10}$?

Page 8, line 27: What do you mean by "artificial area"?

Page 8, lines 28 and 31: EPER data are only available for the EU-countries + Norway — how did you treat industrial emissions in the other countries?

Page 8, line 31: What is "artificial landuse"?

Page 8, lines 36–38: Considering the great uncertainties in the residential combustion emissions I suggest that you give some more details about the emissions you have used in EURODELTA. The statement that "Germany, Sweden, and Spain clearly have the lowest (levels of) emissions" is not clear enough. Do you mean the lowest emission per capita? Or per square km? In order for the results from the EURODELTA modelling to be comparable to other studies I suggest that you add a table to the Supplement specifying annual total national residential combustion emissions assumed in the EURODELTA inventory.

Page 9, line 3: What are "the usual default profiles"?

Page 9, lines 4–6: "a PM speciation profile provided by IIASA (Personal Communication from IIASA) was used to estimate the fraction of Non-carbonaceous species, Elemental Carbon and Organic Matter per activity sectors and country"
- This PM speciation profile must be provided with the article. Personal communication with an organisation (IIASA) is not a reference that makes it possible for readers to find

the relevant information to be able to reproduce the work. A table specifying the three PM$_{2.5}$ and coarse PM fractions for each emission sector and country should be added to the Supplement of the paper.

Page 9, lines 22–23: There is no reference to a description of the SMOKE system.

Page 10, lines 10–15, Wildfire emissions: Which emitted species were included for wildfires? What gases and which particulate species were included (include information about how the PM-emissions were split between organics, BC, and other PM-components)?

Page 10, lines 22–23: Why were the agricultural and road dust PM sources not activated in the LOTOS-EUROS model?

Page 15, lines 4–6, Regarding the PBL and the LOTOS-EUROS and EMEP models: "LOTOS-EUROS and EMEP that should adopt IFS PBL too, show partially different performance, suggesting that the latter models partially recomputed boundary layer height."
- This is too unclear! You have to be able to describe how these models handle the PBL! In what way do they "partially recompute" the BL height?

Page 16, lines 5–6: "The large positive bias in 2007 and negative in 2009 are largely explained by the boundary conditions that are biased respectively of +8 and -20 µg m$^{-3}$"
- I agree about the negative bias in the 2009 campaign but the bias of +8 µg m$^{-3}$ in the 2007 campaign can hardly be considered to "largely explain" the very large positive bias (21–23 µg m$^{-3}$) for CAMx, CMAQ and Chimere — I guess there must be other factors that are more important than the boundary conditions to explain the poor performance of these three models?

Page 16, lines 7–8: "For the summertime campaign 2006 CHIMERE and CMAQ display the lowest correlation for daily averaged concentrations"
- Can you explain the very poor correlation for Chimere and CMAQ for this summer period?

Page 16, lines 11–13: "All models simulate high ozone concentrations over the Mediterranean sea, most of them behaves satisfactorily in Malta and Cyprus stations confirming the ozone concentrations pattern over the seas for the "ensemble" shown in Fig. 6."
- What do you mean by "confirming the ozone pattern over the seas"? Do you mean that a "satisfactory" behaviour at two sites in the Mediterranean region proves that the model ensemble gives good ozone concentrations over all sea areas? Also, in Fig. 6 I see no observation data from Cyprus so for this summer period it is really only one site you base your statement on?

Page 16, lines 20–21: "This result confirms that during stable conditions the pollutant concentration is influenced not only by the PBL height, but also by the overall reconstruction of vertical dispersion."

- What do you mean by "the overall reconstruction of vertical dispersion"? And could the differences of the results not also be due to differences in dry deposition and chemistry?

Page 16, lines 26–28: "Not only the bias is affected by global boundary conditions, but also this result indicates that biased ozone boundary conditions globally impair the normalized statistics confirming the non linearity of ozone chemistry."
- This sentence hardly makes any sense at all to me. I think it is unclear what you mean and it seems like just speculation to me.
- What do you mean by "globally impairing normalized statistics" and how does this "confirm the non linearity of ozone chemistry"?
- As mentioned above I do not think that you have shown that the global boundary conditions is the main reason for the model problems for the 2007 campaign! Of the four ENSEMBLE models Chimere performs very poorly for 2007 (or at least very differently than the other three models) and this can not be explained by the global boundary conditions.

Page 17, lines 8–9: "This underestimation of $NO_2$ concentrations is certainly related to rather high ozone concentrations."
- Can you explain why CAMx behaves differently than the other models (e.g. CMAQ also has high ozone concentrations)?

Page 17, lines 16–17: "Over lands the $NO_2$ chemistry and the different biogenic NO emissions explain a large part of the differences far from urban areas."
- How does this explain the differences between the models — be specific.

Page 17, lines 19–20: "It should be pointed out that the observed $NO_2$ concentrations can be slightly overestimated because of sampling artefact (evaporation of nitric acid)."
- What do you mean by slightly? Give some number/estimate! How large overestimation of $NO_2$ could you possibly get from the evaporation of $HNO_3$?
- Provide a reference for this sampling artefact.

Page 17, lines 32–33: "Differently, differences in diurnal temperature between CMAQ and other models seem less relevant with respect to pollutant concentration."
- How do you know that the temperature differences are less relevant? And does this statement only refer to the $NO_2$-concentrations or to all pollutants?

Page 18, Sect 6.3 Sulphur dioxide
- General comment: This section is very short and essentially only states that the model results for $SO_2$ are quite poor with hardly any explanation why. I think a much more detailed investigation of the differences in deposition and chemistry are needed here.

Page 18, lines 6–7: "The overestimation of the first group of models could be explained as follows for MINNI which has the lowest PBL and RCG having the lowest wind speed."

- The sentence is strangely formulated — perhaps it could have been written something like: "The overestimation in the MINNI model could possibly be partially explained by the low model PBL height"
- However, I do not think that the "explanations" are very satisfying — in my opinion they are not really explanations at all:
  - For 2006 the EMEP model also severely underestimate the PBL height without overestimating $SO_2$.
  - The wind speed in CMAQ is as low as in the RCG model, without overestimation of $SO_2$, and these models actually have the smallest bias for U10 for the 2009 period.

Page 18, lines 20–21, Regarding the CMAQ-results:
- I do not think that the CMAQ results are very different for "at least three campaigns" — it strongly deviates for 2006 and deviates somewhat for 2008 but for the other two campaigns the CMAQ results look "similar" to the other models (at least for the RMSE, which is what was discussed here).

Page 22, lines 6–8, Regarding the $NO_2$ results at the German sites; only meteorological aspects are discussed here, but other things can also lead to modelling problems:
- How do the model results for ozone look at the same sites?
- Could $NO_2$ emissions be underestimated?

Page 22, lines 8–14, regarding the $NO_2$ results in the Po Valley
- Are you sure that you are not having problems with underestimated NOx emissions in this region?

Page 22, lines 26–30, the discussion about the correlation between the performances of the ensemble (RMSE) with the variability of the models is a bit confusing
- What values are you correlating?
- Can low correlation coefficients (-0.2 to -0.3) for only three of four campaigns and only two species be considered significant? What the correlation coefficients for the other species?
- Providing a table with the correlation coefficients for the different species and seasons may could probably make this easier to understand.

Page 23, line 21: What do you mean by "a relevant spatial variability"?

Page 23, lines 25–26: "Such spread can be considered as a measure of the uncertainty related do vertical mixing and qualitatively correspond to 80-100% of the observed mean concentration."
- I do not understand how the model spread can be considered a measure of the uncertainty related to vertical mixing. Could there not be other differences between the models that are important?

Page 23, lines 31–32: As pointed out above I do not think that you have shown that the "lower PBL heights (for MINNI) and wind speed (for RCG)" really **explain** the errors. Also the CMAQ wind speed seems to be as low as the RCG wind speed (according to S0).

Page 24, line 12: "while EMEP seems more able to capture the evolution of the single PM compound."
- Which single PM compound?

Page 24, lines 21–22: "The analysis of individual compounds of PM will bring more detailed, it will be investigated in a companion paper."
- Excluding this detailed information from the present paper makes the whole discussion of PM totally uninteresting.

**Language**

The manuscript is not very well written, which makes it tedious to read. Large parts of the manuscript needs language editing/corrections. It is not the job of the referees of a paper to correct the language — so I only give some examples below, in the Technical corrections section. Some of the 36 authors of the paper are likely very good at English and, since all authors must have seen the manuscript before submission (according to the obligations for authors), I am surprised that they have accepted the submission without helping to improve the language before the paper was submitted. Please make sure that the whole manuscript is checked carefully if it is resubmitted.

**Technical corrections**

Page 1 line 37: "period" → "periods"

Page 1 line 38: "allowing evaluating the influence" → "allowing evaluation of the influence"

Page 2 line 5: "good very similar" do you mean "good and very similar"?

Page 2 line 18: replace "modelling, techniques" by "modelling techniques"

Page 2 line 19: "calculation uncertainty" do you mean "model (or perhaps modelling) uncertainty"?

Page 3 line 7: "exercise" → "exercises"

Page 3 lines 23–24: I guess the list of "non-model" institutes should include NILU as well (since W. Aas is included in the author list)?

Page 3 line 28: replace "join analysis" by "joint analysis"

Page 8, line 36: replace "most of countries" with "most countries" or "most of the countries"

Page 9, line 32: The first sentence of the "Sea salt emissions" paragraph is strange. As formulated it does not make sense.

Page 11, lines 1–2: "was diagnosed in ECMWF was made available" should probably be "as diagnosed in the IFS-ECMWF model was made available"

Page 12, line 12: "most of species" → "most of the species"

Page 12, line 19: "at some EMEP." → "at some EMEP sites."

Page 12, line 27: "converted in m/s" → "converted to m/s"

Page 13, line 1: "Being the boundary layer height a concept valid only for convective" → "Since the boundary layer height is a concept valid only for convective"

Page 13, line 21: "compare" → "compared"

Page 13, line 22: "is" → "was"

Page 13, line 22: "characterized by windy conditions in Europe with cool temperature above average everywhere in Europe" — strange formulation; what do you mean by "cool temperature above average"?

Page 13, line 24–25: "Precipitation were low over the Mediterranean basin but above the climatic average compare to 1961-1990 base period in the rest of Europe." could be changed to "Precipitation was small over the Mediterranean basin but above the climate average, compared to the 1961-1990 period, in the rest of Europe."

Page 13, line 28: "spells end" → "spells in the end"

Page 13, line 28–29: "After some cold spells end of February, March 2009 turned cooler with on average warmer temperatures compare to the 1961-1990 base period" — strange formulation; did March 2009 turn *cooler* than the cold spells in the end of February but it was still *warmer* than the climate average?

Page 14, line 3: "whatever the model" → "for all models"

Page 14, line 6: "this bias exceed" → "this bias exceeds" (or "these biases exceed") and "whatever the campaign" → "for all campaigns"

Page 14, lines 25–26: "In the IFS only 10m winds are used from ships over the oceans for data assimilation (problem of station representativeness for inland stations)." — awkward formulation

— I would suggest something like: "In the IFS only 10m winds from ocean going ships are used in the data assimilation due to problems with station representativity for inland sites."

Page 14, lines 27–29: "For the lowest winds generally observed during nightime the comparison of the predicted diurnal cycle with observations show a largest positive bias at night than during the afternoon (Fig. 2), this behaviour could lead to an overestimation of the advection process." This is a very strange sentence that I do not understand. It needs to be reformulated.

Page 15, line 13: "convention" → "convection"

Page 15, line 17: "use the PBL from ECMWF PBL" → "use the PBL from IFS"

Page 15, line 22: "the negative bias of MINNI has the same order of magnitude as the other models" → "the negative bias of MINNI is of the same order of magnitude as those of the other models"

Page 15, line 23: "are still lower" → "are somewhat lower"

Page 15, line 25: "model" → "models"

Page 15, line 28: "on emission areas" → "in emission areas" and "Besides of urban areas" → "Besides in urban areas" (or perhaps "Besides urban areas")

Page 15, line 29: "that are related to the differences of PBL predicted" → "which is related to the differences in the PBL predicted"

Page 17, lines 14–15: "the mixing of close to emissions is responsible for model output differences" — I think the whole sentence is a bit awkwardly formulated, perhaps this part could be changed to something like: "variations in the PBL height between different models may lead to large differences in modelled concentrations in high-emission areas"

Page 17, lines 32–33: "Differently, differences in diurnal temperature..." — strangely formulated sentence.

Page 18, line 8: "in-deep" → "in-depth"

Page 18, line 9: "This involves a positive bias" → "This leads to a positive bias"

Page 19, line 2: "of the seas" → "over the seas"

Page 19, line 16: "and a few" → "and a little" (or perhaps "and some")

Page 19, line 27: "all models underestimate" → "all other models tend to underestimate"

Page 19, line 31: "Whatever the campaign" → "For all campaigns"

Page 20, line 13: "are coherent with the completeness of our inventory" — I think a better formulation could be "are consistent with our incomplete inventory"

Page 21, line 22: "smaller areas" → "limited areas"

Page 21, lines 25–26: Remove the sentence: "Finally, as already mentioned, PBL heights derived at SIRTA site has been included too." — this manuscript is too long to state this twice within the same paragraph.

Page 22, line 27: "close between" → "close to"

Page 23, line 8: "mainly driven by a relevant underestimation" → "at least partly driven by a major underestimation"

Page 23, line 9: "CTMs are affordable in reproducing ozone" → "CTMs are able to reproduce ozone"

Page 23, line 11: "nigh-time" → "night-time"

Page 23, line 22: "Likewise ozone" → "Similar to ozone" or "As for ozone"

Page 24, line 1: "rely in chemistry" → "be due to chemistry"

Page 24, line 7: "Differently, the RMSE rises up 15 µg m$^{-3}$, representing more than 80% of the observed mean." — incomplete sentence; I guess you mean "rises up to 15 µg m$^{-3}$ for the campaign XXXX..."?

Page 24, line 27: "are still missing in state of art CTM" → "are still missing in some state of the art CTMs"

---

## Referee Comment (RC2) · Anonymous Referee #4 · 16 Jun 2016

General comments

This manuscript is a thorough description of an international model inter-comparison exercise. It can in my view be published in ACP, provided that the comments and concerns below will be taken into account.

The article contains interesting and useful results. However, in my view the discussion of results should focus much more on the results that have some general interest, and on the more general insights and conclusions, and the amount of small details should be substantially reduced. By small details I mean e.g. discussion on how each

individual model has performed for each pollutant and each campaign. The amount of figures and tables is also very large; I would advise the authors to reduce these. However, the figures that that make it possible to draw general conclusions should be included.

I suggest that the authors would add to conclusions a discussion on the most important improvements of the models, and areas of improvement for the CTM's in general in the future, based on their findings. The terminology also should be more precise, and some of the conclusion more cautious, taking into account the limitations of the data; details are discussed below.

Specific comments

Abstract. Explain which experimental datasets were used, and how many stations were included, please. 'Background stations', specify which background; probably regional background, not urban or global background. The discussion would be in my view more clear, if the evaluation of met parameters would be presented first, then evaluation of concentrations. 'performances were good', specify what is meant with 'performance', do you mean e.g. bias or correlations, or both ? PM, specify which PM fraction.

Introduction. In discussing model inter-comparisons, refer also to the most recent relevant ones, especially Prank et al, 2016, ACP (16, 6041–6070). "... showed better performance but higher uncertainty...' define what is meant with 'performance' and what you mean with 'uncertainty'. The institutes participating... this sentence should be deleted; not scientifically relevant information. 'criteria pollutants': define concept (which criteria ? defined by whom ?); probably the authors refer to the latest EU directives or limit values (?); but that should then be specified.

methods. p 8 'lowest levels of emissions': emissions of which pollutant ?

discussion. p 23: 'model formulation and setup ... more influencing than met conditions'. Define what is meant with 'model formulation and set-up' (is it the setup of

input data, which ones ? set-up of model parameters and submodels, which ones ?; or selection of CTM's themselves ?). This statement is also over-interpretation; it has only been shown to be valid for the range of met parameters that were included in the selected conditions, which was not especially wide. Please re-write this, allowing for the limitations of the data used. p. 23. 'highest errors': which stat. model evaluation parameter is meant by 'error' ?

Technical corrections

I would also suggest that the whole text and the language will be checked, and the fairly numerous misprints and language mistakes will be corrected.

---

## Author Comment (AC1) · 19 Aug 2016

Dear Colleague,

Please find enclosed (as supplement document) the answers to your questions and comments. Thank you again for this very complete review. The zip file contains : (i) the reply document, (ii) the revised version with track changes, (iii) the revised version without track changes, (iv) the new supplementary material. The revised version takes into account the corrections suggested by both reviews.

Best Regards,

[Figure]

Bertrand Bessagnet

Please also note the supplement to this comment:
http://www.atmos-chem-phys-discuss.net/acp-2015-736/acp-2015-736-AC1-
supplement.zip

---

## Author Response (AR1)

Referee report regarding the manuscript:

Presentation of the EURODELTA III intercomparison exercise - Evaluation of the chemistry transport models performance on criteria pollutants and joint analysis with meteorology

Authors: B. Bessagnet et al.

**We thank a lot the reviewer for a very complete review of our paper. The answers are written here below in bold characters after each comment.**

General comments

In my opinion this paper is not suitable for ACP — it is essentially a technical report from the EURODELTA project, and I think it should be published as such; a (rather confusingly presented) model intercomparison, for a limited set of "standard" atmospheric components, is not interesting enough for publication in ACP. Many similar model intercomparisons have been published and, as written, this one does not contribute anything new.

**As mentioned in the introduction: "Differently to the previous inter-comparison exercises, most of models have been run in EURODELTAIII with the same input data (emissions, meteorology, boundary conditions) and over the same domain (domain extension and resolution) with some rare exceptions. Participating models were applied over four different periods, within a rather limited number of years thus allowing to evaluate the influence of different meteorological conditions on model performances."**

**The joint analysis of meteorology and criteria pollutants is not so frequent in the literature; we think it is the first time in the frame of an intercomparison exercise such an analysis is performed. For instance, the analysis of the boundary layer height and wind speed of each model are particularly interesting and show how the models are dependent on variables that are not so frequently evaluated.**

The paper only documents model results for seven different chemical transport models without enough detail to be able to draw any useful conclusions for the general scientific community. I can not see what this paper contributes to the understanding about the atmospheric chemistry or physics, or any new information that aids in improving modelling of the atmospheric composition.

**See above. Moreover, as mentioned in the introduction we remind here the objectives of the paper : "The objective of this paper is twofold, (i) to introduce the exercise, the input data and the participating models, and (ii) to analyse the behaviour of models in the four campaigns focussing on the criteria pollutants $PM_{10}$, $PM_{2.5}$, $O_3$, $NO_2$ and $SO_2$ and relevant meteorological variables. Complementary analyses of depositions fluxes and PM composition data at high temporal**

**resolution will be discussed in companion papers in order to better understand the behaviour of models. »**

I also think that the authors are fragmenting their research (over several papers). This should be avoided (see the ACP Obligations for authors). In order to better understand model performance you need to take into account all important processes — including deposition and chemistry. If the authors want to publish this material in a scientific journal I think it has to be combined with the information about deposition and chemical composition of particulate matter. Splitting the model evaluation into three different papers is not appropriate.

**The two objectives of the paper (mentioned before) are not fragmented. It is common to break down the results analysis of such projects in several papers.**

The paper could have been acceptable for Geoscientific Model Development (GMD, which accepts model evaluation papers) — if the presentation had been better — but I think that a much more scientific approach is needed to make the material presented in this manuscript interesting enough for publication in a peer-reviewed journal.

**For the first part of the comment, the editor will also have an opinion. However, we see several model evaluation papers in ACP such as these recent ones:**

**Pan, X., Chin, M., Gautam, R., Bian, H., Kim, D., Colarco, P. R., Diehl, T. L., Takemura, T., Pozzoli, L., Tsigaridis, K., Bauer, S., and Bellouin, N.: A multi-model evaluation of aerosols over South Asia: common problems and possible causes, Atmos. Chem. Phys., 15, 5903-5928, doi:10.5194/acp-15-5903-2015, 2015.**

**Prank, M., Sofiev, M., Tsyro, S., Hendriks, C., Semeena, V., Vazhappilly Francis, X., Butler, T., Denier van der Gon, H., Friedrich, R., Hendricks, J., Kong, X., Lawrence, M., Righi, M., Samaras, Z., Sausen, R., Kukkonen, J., and Sokhi, R.: Evaluation of the performance of four chemical transport models in predicting the aerosol chemical composition in Europe in 2005, Atmos. Chem. Phys., 16, 6041-6070, doi:10.5194/acp-16-6041-2016, 2016.**

**For us, GMD should be more dedicated to model development that is not strictly the topic of our paper here.**

**For the second part of the comment, see above our first answers.**

I am sure that there are a number of interesting scientific questions that the EURODELTA project can answer, and I suggest that the authors focus mainly on that, and keep this kind of model intercomparison/evaluation documentation to technical reports. This paper in itself has little significance for the ACP community.

**As we previously mentioned, we think this paper is the first to address both an analysis on meteorology and chemistry. Most of the previous exercises addressed only comparisons on chemical compounds without any analysis on meteorology. Also, this paper is an introduction of the exercise with other papers that are in preparation or submitted.**

The paper has a very long author list with 36 authors! However, the brief statement on page 3, about what seven of the participating institutes (and I think NILU is missing in this list) have contributed to the project, is not motivation enough for the inclusion of so many authors.

**NILU is included in the list of authors but we forgot it in page 3. Finally, we removed the sentence mentioning the role of partners, following the recommendation of the other referee.**

Considering the very long author list, please give a brief explanation of what each individual has contributed to this paper in the reply to this referee comment (the statement of contributions can be added to the supplement of the paper). Please note Point 9 under the General Obligations for Authors for ACP (my highlighting):

"To protect the integrity of authorship, only persons who have significantly contributed to the research and paper preparation should be listed as authors. The corresponding author attests to the fact that any others named as authors have seen the final version of the paper and have agreed to its submission for publication." … "The author who submits a manuscript for publication accepts the responsibility of having included as co-authors all persons that are appropriate and none that are inappropriate."

**Regarding the number of authors: EURODELTA is an on-going project which started in 2001. This project is a very cooperative project involving an important number of organization and researchers to cover all topics : coordination, data management, modelling, emissions, meteorology, boundary conditions, results analysis, manuscript preparation. For a first paper on a project it is normal to put all people who have participated to the project. As an example, for the MACCII project, we count 60 authors for this paper in GMD (same author policy) with the same number of modeling teams :**

**Marécal, V., Peuch, V.-H., Andersson, C., Andersson, S., Arteta, J., Beekmann, M., Benedictow, A., Bergström, R., Bessagnet, B., Cansado, A., Chéroux, F., Colette, A., Coman, A., Curier, R. L., Denier van der Gon, H. A. C., Drouin, A., Elbern, H., Emili, E., Engelen, R. J., Eskes, H. J., Foret, G., Friese, E., Gauss, M., Giannaros, C., Guth, J., Joly, M., Jaumouillé, E., Josse, B., Kadygrov, N., Kaiser, J. W., Krajsek, K., Kuenen, J., Kumar, U., Liora, N., Lopez, E., Malherbe, L., Martinez, I., Melas, D., Meleux, F., Menut, L., Moinat, P., Morales, T., Parmentier, J., Piacentini, A., Plu, M., Poupkou, A., Queguiner, S., Robertson, L., Rouïl, L., Schaap, M., Segers, A., Sofiev, M., Tarasson, L., Thomas, M., Timmermans, R., Valdebenito, Á., van Velthoven, P., van Versendaal, R., Vira, J., and Ung, A.: A regional air quality forecasting system over Europe: the MACC-II daily ensemble production, Geosci. Model Dev., 8, 2777-2813, doi:10.5194/gmd-8-2777-2015, 2015.**

Specific comments

Page 3, lines 34–35: "As a consequence, there were very limited differences in the models set up, representing a sort of sensitivity analysis to several aspects of the modelling chains."

● I do not understand what you mean by this sentence! What do you mean by "a sort of sensitivity analysis" and in what sense is this study a sensitivity analysis? I suggest that this sentence is removed.

**The reviewer is right, we modified the sentence as: "*As a consequence, most of differences in the outputs will be attributed to the simulation of chemical and physical processes*".**

Page 4, lines 3–4: "Complementary analyses of depositions fluxes and PM composition data at high temporal resolution will be discussed in companion papers in order to better understand the behaviour of models."

● In my opinion this is fragmentation of research papers, and the consequence is that the present paper becomes uninteresting. As mentioned in the General Comments, I do not think that splitting the information this way in three different papers is useful. If knowledge about the deposition fluxes and PM composition data are important for understanding the behaviour of the models (which I certainly expect them to be) this information is needed in the present paper!

**This comment on "fragmentation" is discussed in a previous answer.**

Page 4, lines 25–26: "In CMAQ additional anthropogenic dust is calculated as 90% of unspecified PM coarse emissions and attributed to fugitive dust"

● What is the motivation for adding extra anthropogenic dust? Was this just a modeling mistake? Or do you have good reasons to believe that the emission inventory used in the present study lacks a substantial amount of fugitive dust? And if this is the case, why did you not increase the emissions in all models?

**This is exactly what Binkowski and Roselle (2003) wrote :"The emissions inventory used for this contribution estimates that 90% of PM10 is fugitive dust, and that 70% of this dust consists of PM2.5 particles. The paradigm adopted for the CMAQ model is that fugitive dust is a coarse mode phenomenon with a tail that overlaps the PM2.5 range." This fraction would be the resuspension of dust produced by human activities, the same we investigated in Vautard et al., 2005. We did not switch off this additional emission in CMAQ. Such types of emission parameterization are not available in the other models.**

Page 4, line 27: "CAMx did not activate the sea salts parameterisation in this exercise."

● Why not? Was this a modelling mistake? Or are there problems with the sea salt emissions in CAMx?

**Yes they had problems with the use of the sea salt parameterization. Sea salt modeling in general has large uncertainties mainly in generation of sea spray which occurs as the waves break on the surface of the ocean and whitecaps form. Sea-salt pre-processor of CAMx was not available in the lat-lon grid system at the time of the exercise. The initial attempt to adapt it to the required grid to generate sea salt emissions resulted in too high emissions over the north Atlantic. Due to very high uncertainty, we decided not to include it for this exercise.**

Page 4, lines 30–31: Why was CAMx not included in the ENSEMBLE for O3, NO2, and SO2? I would guess that the lack of sea salt would hardly have any impact on these three gaseous species.

**We preferred to have an ENSEMBLE consistent for all species since we also compared the ENSEMBLE between gases and PM, that's why we excluded CAMx.**

Page 8 — Emissions: Are the emissions used in the EURODELTA project available for use by scientists outside the project? If they are, please specify this and where they can be found. If they are not available, more details are needed regarding the emissions in order for others to be able to evaluate/compare this work to other studies using other emission data. Without more detailed information the work presented in this manuscript can not be considered reproducible.

**The emission data are available, we would suggest to put them on ACP if zip files are permitted, but the amount of data is too large. We wrote that data are available on request: "*The full emission dataset is available on request to INERIS*".**

Page 8, lines 16–22: "EMEP national emissions were kept except for..." 14 countries, for which GAINS emissions were used.

● This seems a bit strange - why did you change emissions for these 14 countries and not for the other countries? Please give a motivation. "Additional factors were applied on two Polish regions (x4 or x8) for PM2.5 and PM10 emissions"

● For which Polish regions? They need to be specified in detail to make this work reproducible. It is also unclear if the x4 factor applies to PM2.5 and x8 to PM10 or if the same factors were used for PM2.5 and PM10?

**Yes it is unclear, we have changed it. The same factor was applied for PM2.5 and PM10, but for two different regions. We rephrased it as : "*The country emissions were re-gridded with coefficients based on population density and French bottom-up data, the methodology (Terrenoire et al., 2015) was extrapolated to the whole Europe. For PM2.5 emissions, the annual EMEP national totals were kept except for the countries: Czech Republic, Bosnia and Herzegovina, Belgium, Belarus, Spain, France, Croatia, Ireland, Lithuania, Luxemburg, Moldavia, Republic of Macedonia, Netherland, Turkey. For these countries, PM2.5 emissions from GAINS were used as this database provides higher numbers and certainly more realistic ones since wood burning is known to be underestimated in the EMEP database (Denier van der Gon et al., 2015). Additional factors were applied on two Polish regions for both PM2.5 and PM10 emissions. As a preliminary solution, domestic combustion emissions from provinces with active coal mines were multiplied by a factor of 8, while those in neighbouring provinces were adjusted by a factor of 4 (Kiesewetter et al., 2015)."***

Page 8, line 27: What do you mean by "artificial area"?

**We mean "built-up" area.**

Page 8, lines 28 and 31: EPER data are only available for the EU-countries + Norway — how did you treat industrial emissions in the other countries?

**They are treated using the "built-up area" proxy**

Page 8, line 31: What is "artificial landuse"?

**We mean "built-up" area. We mentioned it in the revised manuscript.**

Page 8, lines 36–38: Considering the great uncertainties in the residential combustion emissions I suggest that you give some more details about the emissions you have used in EURODELTA. The statement that "Germany, Sweden, and Spain clearly have the lowest (levels of) emissions" is not clear enough. Do you mean the lowest emission per capita? Or per square km? In order for the results from the EURODELTA modelling to be comparable to other studies I suggest that you add a table to the Supplement specifying annual total national residential combustion emissions assumed in the EURODELTA inventory.

**We mean the lowest emission for the whole country, we clarified it. We added a table for the PM emission of sector 2 in the supplementary material S8: "*Residential emissions of particulate matter are dominant in wintertime. In most countries, they come from wood burning or coal uses. Germany, Sweden, Spain clearly have the lowest levels of PM2.5 emissions for this activity sector. Romania, Poland and France have the highest levels of annual total emissions per country (Terrenoire et al., 2015). For this activity sector, the PM2.5 emissions by components are provided in supplementary material S8.*"**

Page 9, line 3: What are "the usual default profiles"?

**We mean the usual vertical profiles, that are used in all models to redistribute the emissions.**

Page 9, lines 4–6: "a PM speciation profile provided by IIASA (Personal Communication from IIASA) was used to estimate the fraction of Non-carbonaceous species, Elemental Carbon and Organic Matter per activity sectors and country"

● This PM speciation profile must be provided with the article. Personal communication with an organisation (IIASA) is not a reference that makes it possible for readers to find the relevant information to be able to reproduce the work. A table specifying the three PM2.5 and coarse PM fractions for each emission sector and country should be added to the Supplement of the paper.

**In the revised manuscript we added the reference Klimont et al. (2013) that is the most appropriate and often cited in ACP for the PM emissions and split into EC/OM/Other:**

**Klimont, Z., Kupiainen, K., Heyes, C., Cofala, J., Rafaj, P., Höglund-Isaksson, L., Borken, J., Schöpp, W., Winiwarter, W., Purohit, P., Bertok, I., and Sander, R.: ECLIPSE V4a: Global emission data set developed with the GAINS model for the period 2005 to 2050: key features and principal data sources, International Institute for Applied Systems Analysis (IIASA), Schlossplatz 1, 2361 Laxenburg, Austria, 8 pp., available at: http://eccad.sedoo.fr/eccad_extract_interface/JSF/page_login.jsf, 2013.**

Page 9, lines 22–23: There is no reference to a description of the SMOKE system.

**We added the web site address; there is no publication, only a user manual.**

Page 10, lines 10–15, Wildfire emissions: Which emitted species were included for wildfires?

What gases and which particulate species were included (include information about how the PM-emissions were split between organics, BC, and other PM-components)?

**We added the species in the paper. The following compounds have been selected: CO, CH4, NOx, SO2, PM2.5, TPM, OC, BC. We did not include VOC as the split.**

Page 10, lines 22–23: Why were the agricultural and road dust PM sources not activated in the LOTOS-EUROS model?

**This was a decision of the TNO modeling team. This kind of parameterization can be considered not robust enough and too dependent on the meteorological driver. The same opinion is shared by the CHIMERE team, the dust resuspension scheme was fitted with the MM5 soil moisture, this variable was differently diagnosed in WRF or IFS for instance.**

Page 15, lines 4–6, Regarding the PBL and the LOTOS-EUROS and EMEP models: "LOTOSEUROS and EMEP that should adopt IFS PBL too, show partially different performance, suggesting that the latter models partially recomputed boundary layer height."

● This is too unclear! You have to be able to describe how these models handle the PBL! In what way do they "partially recompute" the BL height?

**We clarified it in the revised version. Some variation can occur due to differences in the interpolation processes. For EMEP, as explained in page 6 , line 12, a minimum PBL is assigned, explaining the differences on the diurnal cycle Fig. 3.**

Page 16, lines 5–6: "The large positive bias in 2007 and negative in 2009 are largely explained by the boundary conditions that are biased respectively of +8 and -20 µg m-3"

● I agree about the negative bias in the 2009 campaign but the bias of +8 µg m-3 in the 2007 campaign can hardly be considered to "largely explain" the very large positive bias (21–23 µg m-3) for CAMx, CMAQ and Chimere — I guess there must be other factors that are more important than the boundary conditions to explain the poor performance of these three models?

**Yes for this group of models in 2007 this bias on boundary conditions partly explains the overestimation. In winter these models give the highest values, the chemistry processes are certainly the main reasons. We modified the comment.**

Page 16, lines 7–8: "For the summertime campaign 2006 CHIMERE and CMAQ display the lowest correlation for daily averaged concentrations"

● Can you explain the very poor correlation for Chimere and CMAQ for this summer period?

**The poor correlation is associated to both low spatial and temporal correlations. These models have troubles to estimate the background concentration over the Alpine regions. We improved the comment in the revised manuscript as follows: "*The low correlation for CMAQ and CHIMERE is due to the difficulties to reproduce both spatial patterns and day to day variations.*"**

Page 16, lines 11–13: "All models simulate high ozone concentrations over the Mediterranean sea, most of them behaves satisfactorily in Malta and Cyprus stations confirming the ozone concentrations pattern over the seas for the "ensemble" shown in Fig. 6."

● What do you mean by "confirming the ozone pattern over the seas"? Do you mean that a "satisfactory" behaviour at two sites in the Mediterranean region proves that the model ensemble

gives good ozone concentrations over all sea areas? Also, in Fig. 6 I see no observation data from Cyprus so for this summer period it is really only one site you base your statement on?

**Yes there are two sites even for the summer period (the site is difficult to see in the Figure over Cyprus). But you are right two sites are certainly not sufficient for a "confirmation", we rephrased the comment and added this reference : "*Nolle, N., Ellul, R., Heinrich, G., Güsten, H. (2002) A long-term study of background ozone concentrations in the central Mediterranean—diurnal and seasonal variations on the island of Gozo, Atmospheric Environment, Volume 36, Issue 8, March 2002, Pages 1391-1402*"**

Page 16, lines 20–21: "This result confirms that during stable conditions the pollutant concentration is influenced not only by the PBL height, but also by the overall reconstruction of vertical dispersion."

● What do you mean by "the overall reconstruction of vertical dispersion"? And could the differences of the results not also be due to differences in dry deposition and chemistry?

**The reviewer is right, certainly this point must be addressed for a pollutant like NO2 less influenced by chemistry. We have the diurnal profiles for CO and it is coherent with our comment. Then we removed the sentence.**

Page 16, lines 26–28: "Not only the bias is affected by global boundary conditions, but also this result indicates that biased ozone boundary conditions globally impair the normalized statistics confirming the non linearity of ozone chemistry."

● This sentence hardly makes any sense at all to me. I think it is unclear what you mean and it seems like just speculation to me.

● What do you mean by "globally impairing normalized statistics" and how does this "confirm the non linearity of ozone chemistry"?

● As mentioned above I do not think that you have shown that the global boundary conditions is the main reason for the model problems for the 2007 campaign! Of the four ENSEMBLE models Chimere performs very poorly for 2007 (or at least very differently than the other three models) and this can not be explained by the global boundary conditions.

**Yes we agreed and removed this sentence that was unclear.**

Page 17, lines 8–9: "This underestimation of NO2 concentrations is certainly related to rather high ozone concentrations."

● Can you explain why CAMx behaves differently than the other models (e.g. CMAQ also has high ozone concentrations)?

**The reviewer is right, this comment has to be complemented by the previous remark of the reviewer on ozone (Page 16, lines 20–21). Looking at elemental carbon (primary species) in Bessagnet et al. (2014) confirms the hypothesis of an impact of vertical mixing that is different and the minimum Kz quite high in CAMX explain the height dilution of primary compounds. We write in the revised version: "*Bessagnet et al. (2014) showed rather low concentrations of elemental carbon compared to other models, this inert species is particularly sensitive to vertical mixing and***

*CAMx presents the highest minimum diffusion coefficient that is of major importance during stable conditions and partly explaining the low NO2 concentrations."*

Page 17, lines 16–17: "Over lands the NO2 chemistry and the different biogenic NO emissions explain a large part of the differences far from urban areas."

● How does this explain the differences between the models — be specific.

**Far from the anthropogenic sources, the chemical processes and the biogenic emissions have more impact with respect to anthropogenic emissions. We changed to "***Over land the NO2 chemistry and the different biogenic NO emission modules in the models are believed to explain a large part of the differences on NO2 concentrations far from urban areas***".**

Page 17, lines 19–20: "It should be pointed out that the observed NO2 concentrations can be slightly overestimated because of sampling artefact (evaporation of nitric acid)."

● What do you mean by slightly? Give some number/estimate! How large overestimation of NO2 could you possibly get from the evaporation of HNO3?

● Provide a reference for this sampling artefact.

**We added this explanation with a correction:"***For some types of analyzers, NO2 is catalytically converted to NO on a heated molybdenum surface and subsequently measured by chemiluminescence after reaction with ozone. The drawback of this technique is that other oxidized nitrogen compounds such as peroxyacetyl nitrate and nitric acid are also partly converted to NO (Steinbacher et al., 2007)***". The reference is given below:**

**Steinbacher, M., C. Zellweger, B. Schwarzenbach, S. Bugmann, B. Buchmann, C. Ordonez, A. S. H. Prevot,and C. Hueglin (2007), Nitrogen oxide measurements at rural sites in Switzerland: Bias of conventional measurement techniques, J. Geophys. Res., 112, D11307, doi:10.1029/2006JD007971.**

Page 17, lines 32–33: "Differently, differences in diurnal temperature between CMAQ and other models seem less relevant with respect to pollutant concentration."

● How do you know that the temperature differences are less relevant? And does this statement only refer to the NO2-concentrations or to all pollutants?

**The reviewer is right, actually, this statement is not relevant, because CMAQ uses a very different meteorology compared to the others. We therefore removed this sentence and focused on the other models particularly those which reported CO concentrations.**

Page 18, Sect 6.3 Sulphur dioxide

● General comment: This section is very short and essentially only states that the model results for SO2 are quite poor with hardly any explanation why. I think a much more detailed investigation of the differences in deposition and chemistry are needed here.

**This part will be more detailed in a companion paper submitted in Atmospheric Environment in June 2016: "Garcia Vivanco et al., Joint analysis of deposition fluxes and atmospheric**

**concentrations predicted by six chemistry transport models in the frame of the EURODELTAIII project".**

Page 18, lines 6–7: "The overestimation of the first group of models could be explained as follows for MINNI which has the lowest PBL and RCG having the lowest wind speed."

● The sentence is strangely formulated — perhaps it could have been written something like: "The overestimation in the MINNI model could possibly be partially explained by the low model PBL height"

**Yes, thanks, we modified it as suggested.**

● However, I do not think that the "explanations" are very satisfying — in my opinion they are not really explanations at all:

○ For 2006 the EMEP model also severely underestimate the PBL height without overestimating SO2.

○ The wind speed in CMAQ is as low as in the RCG model, without overestimation of SO2, and these models actually have the smallest bias for U10 for the 2009 period.

**We agree that for SO2 it is much more difficult to interpret the model outputs without information on chemistry and deposition fluxes. The impact of the PBL for MINNI is discussed latter in section 7.1 and we can see the "partial" impact. We rephrased the section on SO2 as :**

*"The correlations are rather low for all models in the range 0.2-0.4 for the 2006 campaign to 0.5-0.6 for the 2007 campaign (Fig. 4 and supplementary material S1 for all statistics). Two groups of models are identified CAMx, MINNI and RCG that largely overestimate the concentrations and CHIMERE, CMAQ, EMEP and LOTOS-EUROS which are closer to the observations on average with the best performances on the RMSE. The overestimation in the MINNI model could be partially explained by the low model PBL height. For CAMx, the possible reasons such as the vertical distribution of SO2 emissions near the harbours and coastal areas, insufficient conversion to sulfate and too low deposition were discussed in Ciarelli et al. (2016). This leads to a positive bias of the "ensemble" as shown in Fig. 10 (supplementary material S4) particularly in Western Europe; the normalized RMSE is frequently above 100% in most part of Europe. The main hot spots are located in the Eastern Europe in addition with high concentrations along the shipping routes. The coefficient of variation is the lowest over emission areas but very high in remote areas like over the oceans far from shipping tracks and over mountain areas. This behaviour, very different from a primary species like CO, is a first indication of the very different way to simulate the SO2 chemistry and deposition processes in the models.*

*The diurnal cycles presented in Fig. 11 show a peak at about 10:00 – 12:00. This peak is coherent with the hourly emission profiles of the industrial sector showing an emission peak at the same hours; however, most of models predict a larger decrease in the afternoon. Only CMAQ for the 2007 campaign captures satisfactorily the diurnal profile."*

**Ciarelli, G., Aksoyoglu, S., Crippa, M., Jimenez, J. L., Nemitz, E., Sellegri, K., Äijälä, M., Carbone, S., Mohr, C., O'Dowd, C., Poulain, L., Baltensperger, U., and Prévôt, A. S. H.: Evaluation of European air**

**quality modelled by CAMx including the volatility basis set scheme, Atmos. Chem. Phys., 16, 10313-10332, 10.5194/acp-16-10313-2016, 2016.**

Page 18, lines 20–21, Regarding the CMAQ-results:

● I do not think that the CMAQ results are very different for "at least three campaigns" — it strongly deviates for 2006 and deviates somewhat for 2008 but for the other two campaigns the CMAQ results look "similar" to the other models (at least for the RMSE, which is what was discussed here).

**In fact the CMAQ is very different in 2006 and 2008, we modified the text accordingly.**

Page 22, lines 6–8, Regarding the NO2 results at the German sites; only meteorological aspects are discussed here, but other things can also lead to modelling problems:

**Yes, we agree with the referee, but the aim of this section was indeed to focus only on the relationship between NO2 and meteorology, that –of course- is not the only possible reason of discrepancy, but surely one of the most relevant ones.**

● How do the model results for ozone look at the same sites?

**As shown in the supporting material, ozone for CAMx during the winter campaign is on average in line with observations.**

● Could NO2 emissions be underestimated?

**What is important here is to compare the different behaviour during day time and night time as well as from day to day. A global underestimation or overestimation could be attributed to the emissions but it is not the subject here. However, for traffic emissions we have added references: "*However, underestimation of NOx emissions cannot be ruled out as depicted in Vaughan et al. (2016) or Chen and Borken-Kleefeld (2016), these works highlight the potential underestimation of NOx traffic emissions*".**

**Vaughan, A. R., Lee, J. D., Misztal, P. K., Metzger, S., Shaw, M. D., Lewis, A. C., Purvis, R. M., Carslaw, D. C., Goldstein, A. H., Hewitt, C., N., Davison, B. D., Beevers, S. D., Karl, T. G. Spatially resolved flux measurements of NOx from London suggest significantly higher emissions than predicted by inventories. Faraday Discussions, DOI: 10.1039/c5fd00170f, 2016.**

**Chen Y. and Borken Kleefeld J.: NOx Emissions from Diesel Passenger Cars Worsen with Age. Environmental Science & Technology, 50 (1). pp. 3327-5851, 2016.**

Page 22, lines 8–14, regarding the NO2 results in the Po Valley

● Are you sure that you are not having problems with underestimated NOx emissions in this region?

**Same comments of before, we analyse here the evolution of the bias, not the bias itself. However, we added the previous comment regarding a possible underestimation.**

Page 22, lines 26–30, the discussion about the correlation between the performances of the ensemble (RMSE) with the variability of the models is a bit confusing.

● What values are you correlating?

**We calculate the correlation of the ensemble RMSE and the coefficient of variation of the ensemble.**

● Can low correlation coefficients (-0.2 to -0.3) for only three of four campaigns and only two species be considered significant? What the correlation coefficients for the other species?

**On average the correlations are very low but this slight negative value for these two compounds is significant.**

● Providing a table with the correlation coefficients for the different species and seasons may could probably make this easier to understand.

**In fact, for the other compounds the correlation is close to zero, we mentioned it, we are not sure it is necessary to create a table.**

Page 23, line 21: What do you mean by "a relevant spatial variability"?

**This sentence was removed, it was unclear.**

Page 23, lines 25–26: "Such spread can be considered as a measure of the uncertainty related do vertical mixing and qualitatively correspond to 80-100% of the observed mean concentration."

● I do not understand how the model spread can be considered a measure of the uncertainty related to vertical mixing. Could there not be other differences between the models that are important?

**We admit that this argumentation is a bit short and valid only for primary species. Here we have to remind the results on CO concentrations that show this high variability over the emission zones. Since all models share the raw meteorological variables and since far from emissions area this variability is low, the only explanation comes from the vertical dispersion (Kz) that is differently diagnosed by the models. Particularly in the first layer this will be crucial. We added a sentence on the role of the first layer height that is connected to the vertical diffusion. We have modified the sentence focusing only on primary species.**

**We also wrote in the revised version :" *Such spread for primary species and particularly for CO can be considered as a measure of the uncertainty related to vertical mixing and qualitatively corresponds to 80-100% of the observed mean concentration. The height of the first level is also very important for the mixing and deposition processes, it ranges from 20 m for CAMx and CHIMERE to 90 m for EMEP. To be more representative of surface concentrations a correction is implemented for models having a coarse first surface layer (LOTOS-EUROS and EMEP). ".***

Page 23, lines 31–32: As pointed out above I do not think that you have shown that the "lower PBL heights (for MINNI) and wind speed (for RCG)" really explain the errors. Also the CMAQ wind speed seems to be as low as the RCG wind speed (according to S0).

**Ok we have removed the sentence, it partly explains but it is too uncertain**

Page 24, line 12: "while EMEP seems more able to capture the evolution of the single PM compound."

● Which single PM compound?

**The PM compounds are the inorganic species : sulfate, nitrate and ammonium, we clarified. This refers to the paper Bessagnet et al. (2014): "*The analysis of each PM compound for the 2009 period (Bessagnet et al., 2014) revealed that MINNI and EMEP were characterized by rather different scores, suggesting that their overall performance is influenced in a different way by both chemistry and meteorology*".**

Page 24, lines 21–22: "The analysis of individual compounds of PM will bring more detailed, it will be investigated in a companion paper."

● Excluding this detailed information from the present paper makes the whole discussion of PM totally uninteresting.

**As we explained at the beginning, the goal of this paper is twofold (i) to present the EURODELTA exercise, the input data and the participating models, and (ii) to analyse the behaviour of models in the four campaigns focusing on the criteria pollutants PM10, PM2.5, O3, NO2 and SO2 and relevant meteorological variables, to our knowledge this has never been addressed in previous papers in a multi model exercise.**

**Language**

The manuscript is not very well written, which makes it tedious to read. Large parts of the manuscript needs language editing/corrections. It is not the job of the referees of a paper to correct the language — so I only give some examples below, in the Technical corrections section. Some of the 36 authors of the paper are likely very good at English and, since all authors must have seen the manuscript before submission (according to the obligations for authors), I am surprised that they have accepted the submission without helping to improve the language before the paper was submitted. Please make sure that the whole manuscript is checked carefully if it is resubmitted.

**We thank the reviewer for his help in improving the quality of the manuscript. We agreed and accepted all the comments here below. Sometimes we added a remark in bold character.**

Technical corrections

Page 1 line 37: "period" → "periods"

Page 1 line 38: "allowing evaluating the influence" → "allowing evaluation of the influence"

Page 2 line 5: "good very similar" do you mean "good and very similar"? **Yes we do**

Page 2 line 18: replace "modelling, techniques" by "modelling techniques"

Page 2 line 19: "calculation uncertainty" do you mean "model (or perhaps modelling) uncertainty"? **yes we do**

Page 3 line 7: "exercise" → "exercises"

Page 3 lines 23–24: I guess the list of "non-model" institutes should include NILU as well (since W. Aas is included in the author list)? **Actually we follow the comment of the second reviewer we decided to remove this sentence.**

Page 3 line 28: replace "join analysis" by "joint analysis"

Page 8, line 36: replace "most of countries" with "most countries" or "most of the countries"

Page 9, line 32: The first sentence of the "Sea salt emissions" paragraph is strange. As formulated it does not make sense. **We have corrected** it

Page 11, lines 1–2: "was diagnosed in ECMWF was made available" should probably be "as diagnosed in the IFS-ECMWF model was made available" **Yes**

Page 12, line 12: "most of species" → "most of the species"

Page 12, line 19: "at some EMEP." → "at some EMEP sites."

Page 12, line 27: "converted in m/s" → "converted to m/s"

Page 13, line 1: "Being the boundary layer height a concept valid only for convective" → "Since the boundary layer height is a concept valid only for convective"

Page 13, line 21: "compare" → "compared"

Page 13, line 22: "is" → "was"

Page 13, line 22: "characterized by windy conditions in Europe with cool temperature above average everywhere in Europe" — strange formulation; what do you mean by "cool temperature above average"? **Yes we have removed "cool"**

Page 13, line 24–25: "Precipitation were low over the Mediterranean basin but above the climatic average compare to 1961-1990 base period in the rest of Europe." could be changed to "Precipitation was small over the Mediterranean basin but above the climate average, compared to the 1961-1990 period, in the rest of Europe." **Yes**

Page 13, line 28: "spells end" → "spells in the end"

Page 13, line 28–29: "After some cold spells end of February, March 2009 turned cooler with on average warmer temperatures compare to the 1961-1990 base period" — strange formulation; did March 2009 turn cooler than the cold spells in the end of February but it was still warmer than the climate average? **Yes "milder" is more appropriated**

Page 14, line 3: "whatever the model" → "for all models"

Page 14, line 6: "this bias exceed" → "this bias exceeds" (or "these biases exceed") and "whatever the campaign" → "for all campaigns"

Page 14, lines 25–26: "In the IFS only 10m winds are used from ships over the oceans for data assimilation (problem of station representativeness for inland stations)." — awkward formulation — I would suggest something like: "In the IFS only 10m winds from ocean going ships are used in the data assimilation due to problems with station representativity for inland sites." **Yes it is a better formulation**

Page 14, lines 27–29: "For the lowest winds generally observed during nightime the comparison of the predicted diurnal cycle with observations show a largest positive bias at night than during the afternoon (Fig. 2), this behaviour could lead to an overestimation of the advection process."

This is a very strange sentence that I do not understand. It needs to be reformulated.

**We changed it as :"For the lowest winds, the comparison of the predicted diurnal cycle with observations shows a larger positive bias at night than during the afternoon (Fig. 1), this behaviour could lead to an overestimation of the advection process in the chemistry transport models"**

Page 15, line 13: "convention" → "convection"

Page 15, line 17: "use the PBL from ECMWF PBL" → "use the PBL from IFS"

Page 15, line 22: "the negative bias of MINNI has the same order of magnitude as the other models" → "the negative bias of MINNI is of the same order of magnitude as those of the other models"

Page 15, line 23: "are still lower" → "are somewhat lower"

Page 15, line 25: "model" → "models"

Page 15, line 28: "on emission areas" → "in emission areas" and "Besides of urban areas" → "Besides in urban areas" (or perhaps "Besides urban areas")

Page 15, line 29: "that are related to the differences of PBL predicted" → "which is related to the differences in the PBL predicted"

Page 17, lines 14–15: "the mixing of close to emissions is responsible for model output differences" — I think the whole sentence is a bit awkwardly formulated, perhaps this part could be changed to something like: "variations in the PBL height between different models may lead to large differences in modelled concentrations in high-emission areas"

**In the revised versions we replaced by "…*the differences of mixing in models over emission areas lead to large differences in modelled concentrations…*"**

Page 17, lines 32–33: "Differently, differences in diurnal temperature..." — strangely formulated sentence. **The sentence has been removed based on a previous comment**.

Page 18, line 8: "in-deep" → "in-depth"

Page 18, line 9: "This involves a positive bias" → "This leads to a positive bias"

Page 19, line 2: "of the seas" → "over the seas"

Page 19, line 16: "and a few" → "and a little" (or perhaps "and some")

Page 19, line 27: "all models underestimate" → "all other models tend to underestimate"

Page 19, line 31: "Whatever the campaign" → "For all campaigns"

Page 20, line 13: "are coherent with the completeness of our inventory" — I think a better formulation could be "are consistent with our incomplete inventory". **We would say better : "*are consistent with the level of the completeness of our inventory*"**

Page 21, line 22: "smaller areas" → "limited areas"

Page 21, lines 25–26: Remove the sentence: "Finally, as already mentioned, PBL heights derived at SIRTA site has been included too." — this manuscript is too long to state this twice within the same paragraph.

Page 22, line 27: "close between" → "close to"

Page 23, line 8: "mainly driven by a relevant underestimation" → "at least partly driven by a major underestimation"

Page 23, line 9: "CTMs are affordable in reproducing ozone" → "CTMs are able to reproduce ozone"

Page 23, line 11: "nigh-time" → "night-time"

Page 23, line 22: "Likewise ozone" → "Similar to ozone" or "As for ozone"

Page 24, line 1: "rely in chemistry" → "be due to chemistry"

Page 24, line 7: "Differently, the RMSE rises up 15 µg m-3, representing more than 80% of the observed mean." — incomplete sentence; I guess you mean "rises up to 15 µg m-3 for the campaign XXXX..."? **Ok, we accept this comment.**

Page 24, line 27: "are still missing in state of art CTM" → "are still missing in some state of the art CTMs"
**We thank a lot the reviewer for this review of our paper. The answers are written here below in bold characters after each comment.**

General comments

This manuscript is a thorough description of an international model inter-comparison exercise. It can in my view be published in ACP, provided that the comments and concerns below will be taken into account.

The article contains interesting and useful results. However, in my view the discussion of results should focus much more on the results that have some general interest, and on the more general insights and conclusions, and the amount of small details should be substantially reduced. By small details I mean e.g. discussion on how each individual model has performed for each pollutant and each campaign. The amount of figures and tables is also very large; I would advise the authors to reduce these.

**In the revised manuscript we have separated the discussion and the conclusion. The conclusion section is then more general and shorter. We understand the concern of the reviewer on the number of figures, however, we did an important effort to keep the most essential figures, we prefer to keep all of the current figures in the manuscript that bring a lot in the discussion, except Fig. 1 we have removed.**

However, the figures that that make it possible to draw general conclusions should be included. I suggest that the authors would add to conclusions a discussion on the most important improvements of the models, and areas of improvement for the CTM's in general in the future, based on their findings. The terminology also should be more precise, and some of the conclusion more cautious, taking into account the limitations of the data; details are discussed below.

**We have modified the last section of the conclusions as follows : «** *The study stresses the importance of emission sources particularly in wintertime, wood burning emissions are likely the most underestimated source, through the missing species called semi-volatile organic compounds. Road traffic emissions could also be underestimated, gasoline and diesel vehicles are both concerned, and more generally all activity sectors involving combustion processes can be concerned. In this study, the importance of meteorological data is highlighted, the difficulties for meteorological models to simulate meteorological variables like wind speed and PBL height during stable conditions can lead to dramatic consequences on air quality modelling. Developments in air quality modelling have not only to focus on processes but also on emissions and meteorological*

*input data. To complement the analysis, companion papers will focus on depositions of sulphur/nitrogen compounds and on the behaviour of models for particulate matter species. This ensemble of analyses will help to prioritize the improvement of air quality models used in the frame of the CLRTAP»*

Specific comments

Abstract. Explain which experimental datasets were used, and how many stations were included, please. 'Background stations', specify which background; probably regional background, not urban or global background. The discussion would be in my view more clear, if the evaluation of met parameters would be presented first, then evaluation of concentrations. 'performances were good', specify what is meant with 'performance', do you mean e.g. bias or correlations, or both ? PM, specify which PM fraction.

**Here is the new abstract :**

**" The EURODELTA III exercise has facilitated a comprehensive inter-comparison and evaluation of chemistry transport model performance. Participating models performed calculations for four one-month periods in different seasons in the years 2006 to 2009, allowing the influence of different meteorological conditions on model performances to be evaluated. The exercise was performed with strict requirements for the input data, with few exceptions. As a consequence, most of differences in the outputs will be attributed to the differences in model formulations of chemical and physical processes. The models were evaluated mainly for background rural stations in Europe. The performance was assessed in terms of bias, root mean square error and correlation with respect to the concentrations of air pollutants ($NO_2$, $O_3$, $SO_2$, PM10 and PM2.5), as well as key meteorological variables. Though most of meteorological parameters were prescribed, some variables like the planetary boundary layer (PBL) height and the vertical diffusion coefficient were derived in the model pre-processors and can partly explain the spread in model results. In general the day time PBL height is underestimated by all models. The largest variability of predicted PBL is observed over the ocean and seas. For ozone, this study shows the importance of proper boundary conditions for accurate model calculations and then on the regime of the gas and particle chemistry. The models show similar and quite good performance for nitrogen dioxide, whereas they struggle to accurately reproduce measured sulphur dioxide concentrations (for which the agreement with observations is the poorest). In general, the models provide a close-to-observations map of particulate matter (PM2.5 and PM10) concentrations over Europe with rather correlations in the range 0.4 – 0.7 and a systematic underestimation reaching -10 µg m$^{-3}$ for PM10. The highest concentrations are much more underestimated particularly in wintertime. Further evaluation of the mean diurnal cycles of PM reveals a general model tendency to overestimate the effect of the PBL height rise on PM levels in the morning, while the intensity of afternoon chemistry leading to formation of secondary species to be underestimated. This results in larger modelled PM diurnal variations than the observations show and this is so for all seasons. The models tend to be too sensitive to the daily variation of the PBL. All in all, in most cases model performances are more influenced by the model set-up than the season. The good representation of temporal evolution of wind speed is most responsible for models' skillfulness in reproducing the daily variability of pollutant concentrations (e.g. the development of peak episodes), while the reconstruction of the PBL diurnal cycle seems to play a larger role in driving the corresponding**

**pollutant diurnal cycle and hence determine the presence of systematic positive and negative biases detectable on daily basis.”**

Introduction. In discussing model inter-comparisons, refer also to the most recent relevant ones, especially Prank et al, 2016, ACP (16, 6041–6070). “. . . showed better performance but higher uncertainty. . .’ define what is meant with ‘performance’ and what you mean with ‘uncertainty’. The institutes participating. . . this sentence should be deleted; not scientifically relevant information. ‘criteria pollutants’: define concept (which criteria ? defined by whom ?); probably the authors refer to the latest EU directives or limit values (?); but that should then be specified.

**Yes, we have included the references and taken into account all the remarks. Criteria pollutants refer to the Air quality directives, we have modified accordingly. Uncertainties are related to the model formulation (parameterization) and input data. We modified it as :”** *The objective of this paper is twofold, (i) to present the exercise, the input data and the participating models, and (ii) to analyse the behaviour of models in the four campaigns focussing on the criteria pollutants PM10, PM2.5, O3, NO2 and SO2 as defined in the EU directive on air quality 2008/50/EC (EC, 2008), and relevant meteorological variables.”*

Methods. p 8 ‘lowest levels of emissions’: emissions of which pollutant ?

**They refer to $PM_{2.5}$ for the residential sector, we clarified it in the revised version.**

Discussion. p 23: ‘model formulation and setup . . . more influencing than met conditions’.

Define what is meant with ‘model formulation and set-up’ (is it the setup of input data, which ones ? set-up of model parameters and submodels, which ones ?;or selection of CTM’s themselves ?). This statement is also over-interpretation; it has only been shown to be valid for the range of met parameters that were included in the selected conditions, which was not especially wide. Please re-write this, allowing for the limitations of the data used.

**We replaced this statement by «** *This confirms once again that on average and for the limited dataset used in this exercise, the model formulation (parameterization of chemical / physical processes, calculation of meteorological diagnosed variables) and set-up (number of vertical levels, value of key parameters, etc...) are more influencing than raw meteorological conditions on model performance.* **».**

p. 23. ‘highest errors’: which stat. model evaluation parameter is meant by ‘error’ ?

**We have replaced by RMSE instead of error.**

Technical corrections

I would also suggest that the whole text and the language will be checked, and the fairly numerous misprints and language mistakes will be corrected.

**Yes, we have revised the language; there were several misprints and mistakes.**

[revised manuscript text omitted]